# Comprehensive characterization of 536 patient-derived xenograft models prioritizes candidates for targeted treatment

Hua Sun [1,2,27], Song Cao[1,2,27], R. Jay Mashl[1,2,27], Chia-Kuei Mo[1,2,27], Simone Zaccaria [3,4,27], Michael C. Wendl[2,5,6], Sherri R. Davies [1], Matthew H. Bailey[7], Tina M. Primeau[1], Jeremy Hoog[1], Jacqueline L. Mudd[1], Dennis A. Dean II[8], Rajesh Patidar[9], Li Chen[9], Matthew A. Wyczalkowski [1,2], Reyka G. Jayasinghe[1,2], Fernanda Martins Rodrigues[1,2], Nadezhda V. Terekhanova[1,2], Yize Li [1,2], Kian-Huat Lim [1,10], Andrea Wang-Gillam[1,10], Brian A. Van Tine [1,10], Cynthia X. Ma [1,10], Rebecca Aft[10], Katherine C. Fuh[1,10], Julie K. Schwarz [10,11], Jose P. Zevallos[10,12], Sidharth V. Puram[10,12], John F. Dipersio [1,10], The NCI PDXNet Consortium*, Brandi Davis-Dusenbery[8], Matthew J. Ellis[13], Michael T. Lewis[13], Michael A. Davies [14], Meenhard Herlyn [15], Bingliang Fang[14], Jack A. Roth [14], Alana L. Welm [7], Bryan E. Welm[7], Funda Meric-Bernstam [14], Feng Chen[1], Ryan C. Fields[10], Shunqiang Li[1,10], Ramaswamy Govindan[1,10], James H. Doroshow[16], Jeffrey A. Moscow[17], Yvonne A. Evrard [9], Jeffrey H. Chuang [18], Benjamin J. Raphael [3] & Li Ding [1,2,6,10✉]

Development of candidate cancer treatments is a resource-intensive process, with the research community continuing to investigate options beyond static genomic characterization. Toward this goal, we have established the genomic landscapes of 536 patient-derived xenograft (PDX) models across 25 cancer types, together with mutation, copy number, fusion, transcriptomic profiles, and NCI-MATCH arms. Compared with human tumors, PDXs typically have higher purity and fit to investigate dynamic driver events and molecular properties via multiple time points from same case PDXs. Here, we report on dynamic genomic landscapes and pharmacogenomic associations, including associations between activating oncogenic events and drugs, correlations between whole-genome duplications and subclone events, and the potential PDX models for NCI-MATCH trials. Lastly, we provide a web portal having comprehensive pan-cancer PDX genomic profiles and source code to facilitate identification of more druggable events and further insights into PDXs' recapitulation of human tumors.

A full list of author affiliations appears at the end of the paper.

Patient-derived disease models have emerged as important platforms for cancer research[1–3]. In particular, patient-derived xenograft models (PDXs), which are composed of immunodeficient mice engrafted with patients' cancerous material, generally offer more faithful representations than cancer cell-lines, which tend to diverge over time[4]. Uses of PDXs in cancer research are myriad, ranging from investigating basic biology, to discovering biomarkers for therapy response and resistance, to conducting translational cancer research[5]. Their application to drug discovery[6] enables pre-clinical evaluation of therapeutic agents and furnishes a platform for exploring novel drug combinations[7]. The possibility of guiding treatment for rapidly proliferating cancers[8] is suggested by their short timeframes. Likewise, their application to co-clinical trials in which the model is treated with the same regimens as the originating patient tumor may allow for further assessment of the accuracy of the PDX response.

Development of candidate cancer treatments is a resource-intensive process and the research community continues to investigate options beyond static genomic characterization. One promising option is seeking actionable molecular alterations without particular regard for the underlying cancer type. This approach is a centerpiece of The National Cancer Institute (NCI) Molecular Analysis for Therapy Choice (NCI-MATCH or EAY131) trial[9,10], in which cancer patients are assigned to sub-protocols according to aberrations in putatively relevant genes or pathways[11]. The rapid growth of PDX resources has resulted in major efforts to catalog PDX models, harmonize metadata, and organize repositories, including the NIH-NCI PDX Development and Trial Centers Research Network (PDXNet, pdxnetwork.org), the NIH-NCI Patient-Derived Models Repository (PDMR, pdmr.cancer.gov), and EurOPDX (europdx.eu). These PDX resources are supported bioinformatically by the PDX Finder web portal (pdxfinder.org)[12] and by the PDX Minimal Information standard guidelines[13] for describing essential aspects of PDX model derivation.

In the spirit of this grand effort, we obtained 2,028 human and PDX tumor samples representing 536 PDX model lines (511 patients) across 25 cancer types to perform systematic PDX genomic characterization. The heterogeneities characteristic of human progenitor tumors (primary, metastatic, or recurrent) make obtaining the mutational landscapes of PDXs essential for assessing model fidelity, identifying the types and subtypes of cancers that can effectively be captured, and determining whether druggable driver mutations are being recovered. Comprehensive PDX characterization is also crucial for identifying where greater representation is needed to bolster statistical power and for revealing cancer types and subtypes for which PDX models may be more difficult to establish, thereby steering efforts toward alternative platforms like organoids. Analysis of PDX tumors presents challenges beyond those of human matched tumor/normal samples, including lack of germline samples and presence of mouse reads, necessitating additional sample quality control pipelines, tumor-only pipelines, and filtering. Through the use of several bioinformatic pipelines, we identified mutational land-scapes, copy number (CN) alterations, cis/trans mutation sta-tuses, gene fusions, and pan-cancer groups, and compared the results to data from The Cancer Genome Atlas (TCGA). We then investigated dynamic tumor evolution via copy number alteration and whole-genome duplication in PDX models with multiple PDX passages. Lastly, we identified PDX models that meet NCI-MATCH study arm criteria and provide the list of genomic alterations. In summary, these analyses comprehensively char-acterize the genomic features of PDX models and serve as a rich resource for identifying potential models for use in conjunction with clinical trials or for testing experimental drug combinations.

## Results

### Xenografts and clinicopathological summary of sequenced samples

The samples used in this study included tumors from human patients, their derivative PDX models and subsequent passages of PDX models (Fig. 1). We collected whole-exome sequencing (WES) and RNA-seq data for human and PDX samples (Fig. 1a) from PDXNet centers and the PDMR, retaining those samples having good coverage and consistent pedigrees among human and PDX samples per our quality-control (QC) assessment (see Methods). The resulting 3,705 WES and RNA-seq data ($n = 2,321$ unique tissue samples) represent 511 patients and 536 PDX models across 25 cancer types (Fig. 1b, c and Supplementary Data 1), with over 85% of the samples associated with breast (BRCA), colorectal (COAD, READ), sarcoma (SARC), lung (LUAD, LUSC, SCLC), pancreatic (PAAD), skin (SKCM), head and neck (HNSC), bladder (BLCA), and kidney (KIRC) cancers (Fig. 1d). We grouped the cases into classes: those with matched human tumor/normal samples ($n = 186$), with either human tumor ($n = 72$) or normal ($n = 107$) but not both, and those with no human samples ($n = 146$). The purpose of these classifications is to identify cases having tumor and PDX samples but lacking a normal, thereby relegating these cases to a pooled tumor approach for somatic variant calling (see Methods).

Around 55% of cases admit two or more unique PDX passages (Fig. 1c), making them suitable for dynamic tumor evolution analysis. Patient samples contributed relatively few RNA-seq data sets, whereas PDX samples contributed comparable numbers of WES and RNA-seq data sets, together representing over 80% of all samples analyzed. Clinical annotations (Fig. 1e) indicate that of the 536 PDX models, 329 were derived from primary, 159 from metastatic, 28 from recurrent, and 20 from other/unknown tumor types. Prior treatment can have a substantial impact on in vivo drug response. Here, 48% of patients are reported to have had drug treatment prior to specimen collection for PDX engraft-ment, while 44% were reported not to have received any. Further, our human cases represent a wide range of age groups, approximately equal numbers of males and females, and at least three ethnicities. Regarding distribution of cases by center (Fig. 1e), half came from PDMR, a quarter from Washington University, and a quarter from PDXNet Patient Development and Trial Centers (PDTCs) at the University of Texas M.D. Anderson Cancer Center (MDACC), Baylor College of Medicine (BCM), Huntsman Cancer Institute (HCI), and the Wistar Institute (WI).

### The landscape of genetic alterations in 268 human tumors and 536 PDX models

We performed a comprehensive analysis of somatic mutations, copy-number alterations (CNAs), and fusion events on tumors from 511 cancer patients yielding 536 PDX models across multiple cancer types to identify key genetic alterations in the xenografts and compare them to the human tumors (Fig. 1d). These models are summarized on the "PDX Variant Viewer" web portal (https://pdx.wustl.edu/pdx), which organizes cancer types, models, corresponding clinical informa-tion (patient age range, self-reported race, and gender, and PDX specimen treatment status), the derived PDX samples, and the type of variant calling pipeline used (tumor-normal or tumor-only). Somatic variants (fusions, CNVs, and mutations) collated by gene can be viewed by navigating the hyperlinks.

Regarding non-synonymous mutations, overall, we found high variant allele fractions (VAFs) in PDXs compared to both human tumors from the current study and from TCGA, with two peaks at ~0.5 and 1 (Fig. 2a, left). For human tumors, the right panels show calculated VAFs for the top 10 frequently mutated hotspots found in the PDX cohort versus TCGA results, where hotspot mutations were limited to 299 cancer genes[14]. We found

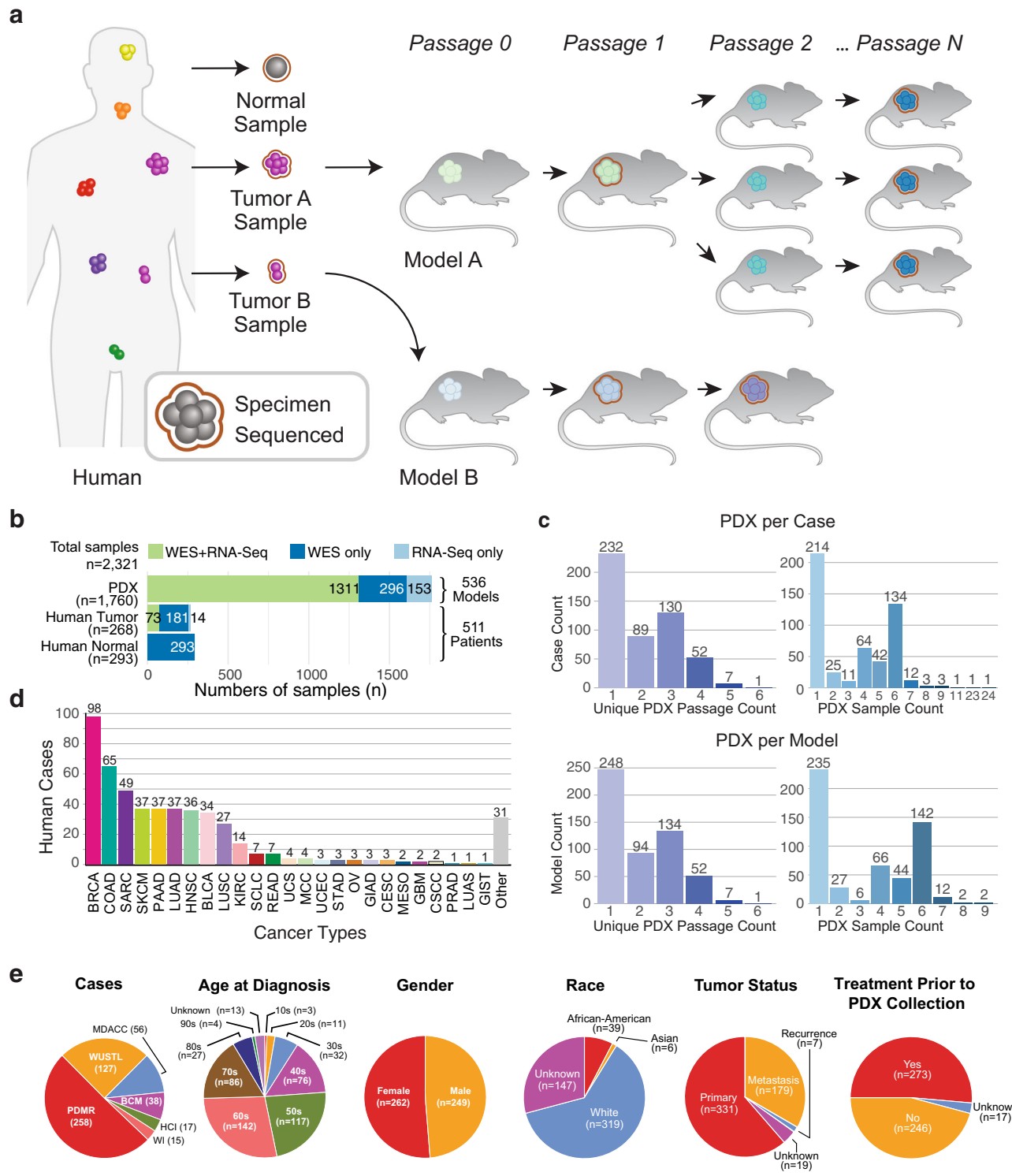

consistently higher VAFs in PDXs for these 10 hotspot mutations ($p < 10^{-4}$, Wilcoxon test), suggesting the PDXs have a high tumor purity. Interestingly, we found that important hotspots R175H, R248Q, R248W, and R273H in *TP53* are homozygous with VAFs close to 1, whereas oncogenic hotspots in *BRAF*, *KRAS*, and *PIK3CA* are heterozygous with VAFs close to 0.5 in PDXs. These trends are not apparent from VAFs of human tumors due to lower purities (Fig. 2a), suggesting the importance of using PDXs to characterize genomic alteration events in tumors more effectively.

We investigated mutations between PDXs and primary human tumors in SKCM, LUSC, SARC, PAAD, LUAD, COADREAD, and BRCA, which each encompass at least 20 patients having mutational data in both PDX and human tumors (Fig. 2b, left). We calculated the mutational similarity (percentage of overlapping mutations) for all mutations and driver-only mutations (Fig. 2b, right), observing that the majority is conserved between PDXs and the corresponding human tumors with a median value of 0.75 across cancer types. However, 27 models (13 from LUAD) showed much lower mutational similarity scores (< 0.2) between human tumor and

**Fig. 1 Data summary. a** Project schematic, showing a human image to represent multiple patients with different cancer diagnoses (pan-cancer). Tumor samples, implanted in mice to form PDX models, are propagated through a sequence of hosts, some of which provide specimens for next-generation sequencing and genomic analysis. **b** Sample types (*left*) and sequence data by assay (*top*) and associated counts of human cases and PDX models (*right*). **c** Distributions of cases according to the number of PDX passage indices (*upper left*) and PDX samples (*upper right*) in the lineage. Analogous distributions for PDX models (*lower set*) are also shown. **d** Distribution of cases by cancer type, following TCGA study abbreviations where possible. **e** Clinical features of cases (age at diagnosis, gender, self-reported race) and specimens (tumor status, treatment status), and PDTC source of sequence data. Key: PDTC, PDX Development and Trial Center; PDX, patient-derived xenograft; RNA-seq, RNA sequencing; TCGA, The Cancer Genome Atlas; WES, whole exome sequencing. Cancer type definitions in this work: BLCA, Bladder/urothelial carcinoma; BRCA, breast carcinoma; CESC, cervical carcinoma; COAD, colon adenocarcinoma; CSCC, cutaneous squamous cell carcinoma; GBM, glioblastoma multiforme; GIAD, gastrointestinal carcinoma, NOS; GIST, gastrointestinal stromal tumor; HNSC, head-and-neck squamous cell carcinoma; KIRC, kidney renal clear cell carcinoma; LUAD, lung adenocarcinoma; LUAS, lung adenosquamous carcinoma; LUSC, lung squamous cell carcinoma; MCC, Merkel cell tumor; MESO, mesothelioma; OV, ovarian carcinoma (epithelial or NOS); PAAD, pancreatic adenocarcinoma; PRAD, prostate carcinoma, NOS; READ, rectal adenocarcinoma; SARC, sarcoma; SCLC, small cell lung carcinoma; SKCM, skin cutaneous melanoma; STAD, stomach adenocarcinoma; UCEC, uterine corpus endometrial carcinoma; UCS, uterine carcinosarcoma. Source data are provided as a Source Data file.

PDXs (Supplementary Fig. 1a). As a result, we saw an over-representation of *EGFR* mutations in these samples. QC via germline variants did not reveal any sample mismatches between PDXs and primary tumors that would explain this low similarity (Supplementary Data 2). Further investigation showed the cause in the majority of models to be that several mutations in human samples had disappeared in PDX models. Subclone selection and purity were important underlying factors. For instance, in model 193523_008_R, human primary tumor had 703 mutations while PDX models only had around 30 mutations. More than 90% of PDX mutations were in the human primary tumor and driver TP53 missense mutation (G245S) was conserved in both the human primary tumor and PDXs, suggesting that PDXs are indeed from the human primary tumors, which is consistent with germline QC. However, only a small subset of mutations appears to be selected during the passaging from human tumor to PDXs, evidently contributing to the loss of mutations in PDX. Another factor is purity, as seen in model MDACC_TC286, in which the PDX has a low purity of 0.28. Here, we only detected 5 mutations, an enormously lower number than that detected in the human tumor (1552). Driver mutations found in human tumors were more conserved between PDXs and human tumors compared to all mutations' results, as demonstrated by a median similarity score of 1 across cancer types (Fig. 2b).

We further examined the mutational similarity for cases with multiple models by calculating both intra- and inter-mutational similarities among different models from the same patient case, where intra cases compare two PDXs derived from the same original patient tumor fragment (i.e., same model) and inter cases compare PDXs derived from different tumor fragments from the same patient (i.e., different models). Figure 2c shows the comparison between inter- and intra-mutational similarities. In general, they are correlated, with intra mutational similarity being higher. Since PDXs from different models originate from different tumor material of the same patient, the low inter mutational similarity suggests intrinsic tumor heterogeneity among tumor segments[15]. Figure 2c shows one example (PDMR-616732) from PAAD, in which the average inter- and intra-mutational similarities are 0.56 and 0.84, respectively. The two models originate from two different metastatic human specimens, collected from liver (R2) and pleura (R3). Both R2 and R3 contain key driver mutations KRAS G12V and TP53 Y235C[16], which are also conserved in their derived PDX passages. The similarity matrix in the right clearly shows a high intra-mutational similarity and a low inter-mutational similarity. We also found high mutational similarity for two samples which are close neighbors in the tree plot. For instance, two PDX passages (PR0 and AK5) from the same parental PDX root (N46) show a highest similarity 0.91.

Most of the patient cases, 98 and 72, were from breast (BRCA) and colorectal (COADREAD) cancer, respectively (Fig. 1d).

Regarding the latter, we combined the colon adenocarcinoma (COAD) and rectum adenocarcinoma (READ) groups to increase statistical power, consistent with TCGA[17]. We then examined genetic alterations in significantly mutated genes (SMGs) established by a large TCGA pan-can study[14] for these cancer types (Supplementary Fig. 1b, c). For BRCA, *TP53* and *PIK3CA* are the two highest mutated genes, again consistent with TCGA (Supplementary Fig. 1b)[18]. We found a higher frequency of *TP53* mutations, which is related to the higher number of basal subtypes included here. In addition to driver mutations, we observed several copy number (CN) deletions in tumor suppressors, such as *TP53*, *PTEN*, *RB1*, and *NF1* in BRCA and CN amplifications in oncogenes (*PIK3CA*, *GATA3*, and *FOXA1*). A few fusions were observed in *PIK3CA* and *MAP3K1*. We found that driver mutations in PDXs appeared to be stable across multiple passages and similar to those in matched primary human tumors (Supplementary Fig. 1b). This finding supports the view that PDXs are representative of their original human tumors at the mutation level. We also found that six patients had pathogenic germline variants in *BRCA1* and *BRCA2*, two genes highly relevant in breast cancer[19]. Our analysis of genetic alterations in COADREAD (Supplementary Fig. 1c) found APC was the highest mutated gene in accordance with the TCGA study[17]. Copy number deletions and amplifications were observed in tumor suppressors *TP53* and *SMAD4* and oncogenes *KRAS*, *PIK3CA*, and *EDNRB*, respectively, consistent with their respective deleterious and activating roles. We also observed the stable evolution of driver mutations with human tumors and across PDX passages, supporting the feasibility of utilizing PDXs to mimic their respective primary human tumors for evaluating drug responses.

**Oncogenic events affected by driver mutations**. We performed an extensive cis and trans study on RNA expression by incorporating TCGA and PDX data to identify biological and clinical relevance of driver mutations in SMGs. We focused on cancer types having over 20 patient PDX models, namely BRCA, COADREAD, SARC, SKCM, PAAD, LUAD, HNSC, BLCA, and LUSC. Figure 3a–c show three selected cancer types with substantial numbers of cis and trans events observed in PDXs. Overall, we found concordance between TCGA and PDX data for key cis and trans events. However, TCGA data encompass many unique events due to a larger sample size, with other factors like sampling also contributing to the difference (Methods). Identification of key oncogenic events from PDXs and the resemblance to human TCGA data suggest candidates for clinical drug trials, as detailed below.

For tumor suppressor genes, we observed a general trend of down-regulated expression in mutated tumors. For instance,

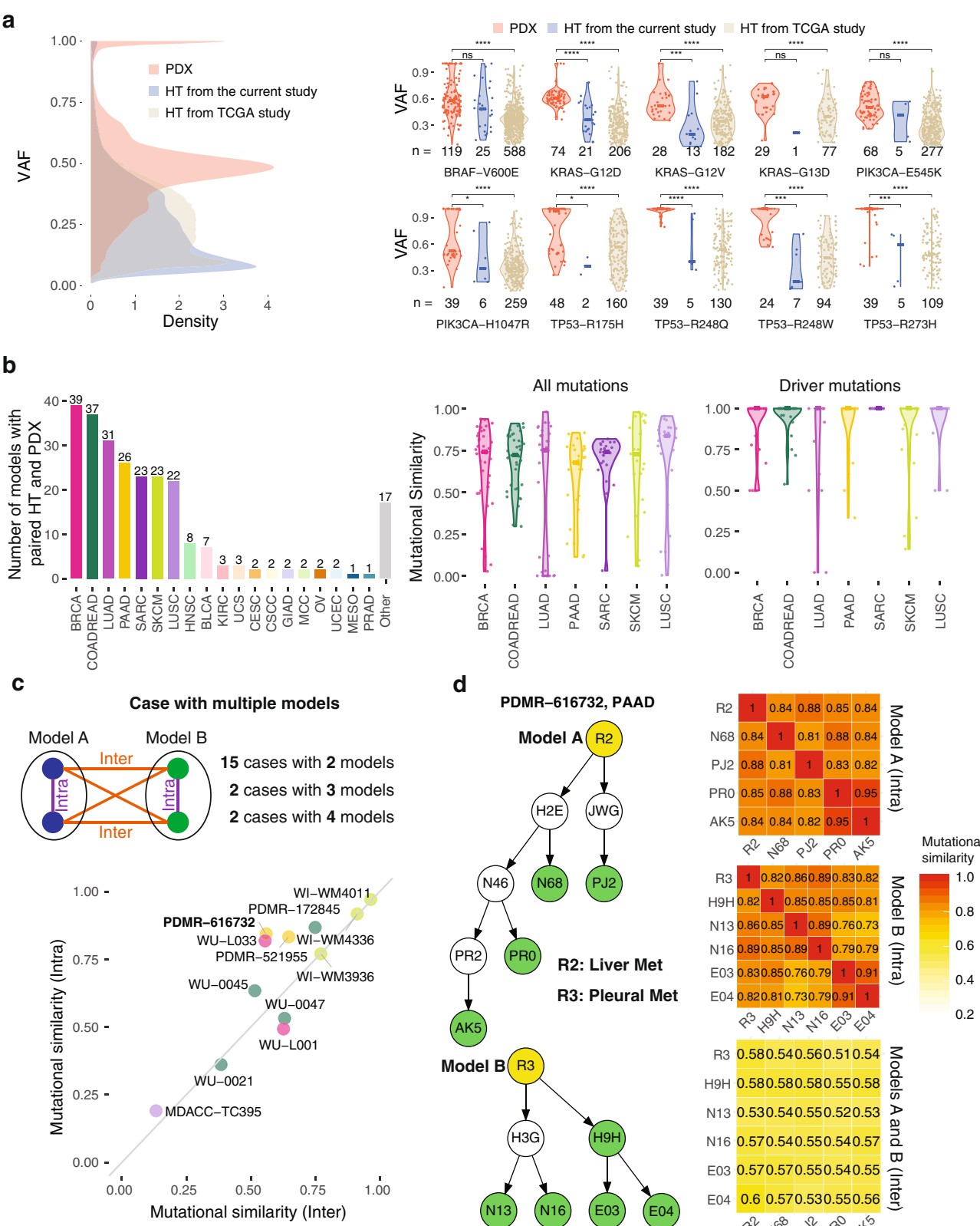

**Fig. 2 The landscape of somatic mutations in human tumors and PDXs. a** Comparison of variant allele frequency (VAFs) for mutations between PDX and TCGA data. Number of samples (n) for each group is shown in the figure. **b** Number of patients with both human and PDX samples and the similarity of somatic mutations between them across different cancer types. **c** Mutational similarity among samples from different PDX models. **d** Schematics of tree plots of two PDX models from example case PDMR-616732 and the heatmap matrix of intra- and inter-mutational similarity. The mutational similarity quantifies the percentage of overlapping mutations between two samples. In a., we use two-sided Wilcoxon rank-sum test for calculating p values, where *, **, ***, and **** stand for p-value < 0.05, < 0.01, < 0.001, and < 0.0001 respectively. Source data are provided as a Source Data file.

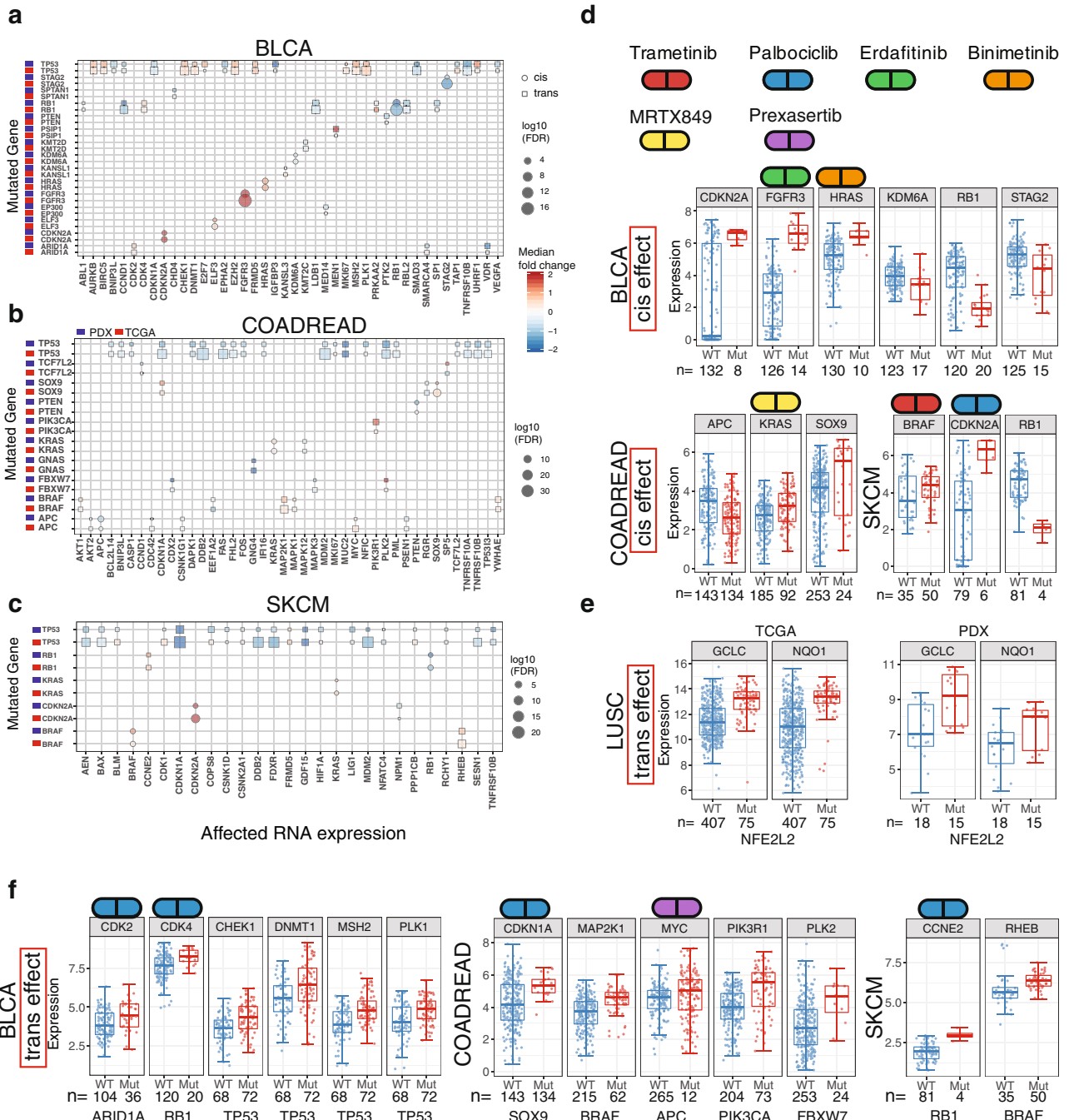

**Fig. 3 Cis and trans effect of driver mutations on gene expression. a–c** Cis and trans effect of driver mutations on RNA expression from TCGA tumors and PDXs on three different cancer types: bladder (BLCA), colorectal (COADREAD), and skin cutaneous melanoma (SKCM). Boxplots for the cis effect (**d**) and trans effect (**e, f**) of key driver genes on gene expression. Number of samples (n) for each group is shown in the figure. The box boundary of each box plot indicates third quartile and first quartile respectively from the top to bottom. The whisker on top were drawn out from the third quartile to the largest data point or up to 1.5 × IQR. Similarly, the bottom whisker extends from the first quartile down to 1.5 × IQR or the lowest data point. The red dot at the center indicates medium. Source data are provided as a Source Data file.

the adenomatous polyposis coli (APC) gene, which is the most frequently mutated gene in COADREAD, displays down-regulated expression in *APC*-mutated samples in both TCGA and PDX data (Fig. 3b, d). A similar trend was also found in phosphatase and tensin homolog (PTEN) in COADREAD and

Stromal Antigen 2 (STAG2) in BLCA. We found a large number of down-regulated cis and trans events in *TP53*-mutated samples in COADREAD, BLCA, and SKCM. A majority of these genes fall into the generic transcription pathway, including *DDB2*, *MDM2*, and *CDKN1A*, which is consistent with the widespread regulation

network by p53 transcription factor[20]. In addition, we found that Retinoblastoma (RB1) was frequently mutated in multiple cancer types, such as BLCA and SKCM (Fig. 3). Based on TCGA data, we observed the down-regulation of RB1 expression, accompanied by increased expression of its interacting partners, such as Cyclin-dependent kinases (CDK), Cyclin E (CCNE), and mini-chromosome maintenance protein complex (MCM) genes (Supplementary Fig. 2). RB1 inhibits MCM2-7 activity through negative feedback[21], with lack thereof in RB1-mutated samples resulting in a high MCM2, MCM4, and MCM6 expression, which is associated with high cell proliferation[22]. Although upregulations of MCM2, MCM4, and MCM6 expression did not attain FDR significance in RB1-mutated xenografts due to small sample size, we indeed observed a trend of increased MCM2 and MCM4 expression in RB1-mutated samples (Supplementary Fig. 2). In addition, we observed high CDK4 expression in both RB1-mutated TCGA and PDX tumors (Fig. 3b,f), suggesting PDX drug trials by CDK inhibitors, like palbociclib.

Driver mutations in BRAF are frequently observed in colorectal and skin cancers[17,23], with a majority falling in the category of V600 hotspot. Figure 3c, d shows high expression in BRAF mutants in both PDX and TCGA tumors, indicating activation events in both cancer types. The observation of high BRAF expression in PDXs suggests drug treatment by FDA-approved BRAF inhibitors, such as Trametinib, Vemurafenib, and Encorafenib. We also observed high MAP2K1 (or MEK1) expression in both PDXs and TCGA tumors in COADREAD, suggesting studying response to a combination of BRAF and MEK inhibitors in BRAF mutated colorectal PDXs[24]. Such has shown favorable outcomes in treating BRAF-mutated melanomas and colorectal cancers, although drug resistance can eventually develop[25,26]. We observed a high Ras homolog enriched in brain (RHEB) expression in BRAF-mutated SKCM (Fig. 3c, f). RHEB is involved in the mTOR pathway via its production of the Rheb protein which binds and regulates mTOR kinase[27]. A recent study has shown that small-molecule NR1 binds and inhibits RHEB[28]. Note that we did not observe up-regulation of RHEB in COADREAD in BRAF-mutated samples in either TCGA or PDX data, suggesting that RHEB upregulation is specific to BRAF-mutated SKCM rather than COADREAD. PDX sample (ID: WM3936-1) with BRAF hotspot mutation p.V600E has also shown good response to BET and MEK inhibitors (OTX+PD901)[29].

We also observed high expression of other oncogenes, including FGFR3 and KRAS in both TCGA and PDX samples (Fig. 3a, b, d), suggesting trials of FGFR or KRAS inhibitors (Erdafitinib or MRTX849) in xenografts with these mutants. Erdafitinib is an FDA approved drug for treating BLCA with FGFR3 mutation and an early clinical study showed that AMG 510 is a potential candidate for treating tumors with KRAS mutations[30]. Good drug response to FGFR or KRAS inhibitors to samples with FGFR or KRAS alteration has been observed in NCI-MATCH study arms (Supplementary Data 5). Driver mutations in oncogene PIK3CA, which encodes the p110α catalytic subunit of PI3 kinase (PI3K), can promote tumor progression by activating the PI3K pathway[31]. Figure 3b shows that PIK3R1, the complex partner of PIK3CA, and not PIK3CA itself, is upregulated in PIK3CA mutated COADREAD tumors from TCGA and PDX data. Drugs specifically targeting the PI3K pathway have shown promising clinical response in PDX models (HCI-003, HCI-013, WHIM12, and WHIM20) with PIK3CA mutations[32–34]. In addition to the PIK3CA and PIK3R1 complex, the other notable interaction network we observed is for genes involved in the NFE2L2 (NRF2) antioxidant signal pathway, which is frequently mutated in LUSC[35]. Driver mutations in NFE2L2 are located in the DLG and ETGE domains, which disrupt the interaction between KEAP1 and NFE2L2, resulting in

the activation of NFE2L2[36]. In LUSC, we observed that genes involved in the NRF2 pathway (GCLC and NQO1) are up-regulated in NFE2L2 mutated samples in both TCGA and PDX data (Fig. 3e), suggesting PDXs recapitulate the key signaling pathway found in human tumors and can serve as an important model system for testing responses to drugs that target a specific signaling pathway. Recent PDX treatment shows good response for GLUT inhibitors for lung cancer PDX models (IDs: TC333, TC453, and TC494) carrying KEAP1 or NFE2L2 mutations[37]. GLUTs play an important role in antioxidant defense[38].

**Oncogenic fusion driver events in PDXs.** Oncogenic kinase fusions with elevated kinase expression could be therapeutic targets for kinase inhibitors[33]. Consistent with a recent TCGA study[39], we observed elevated numbers of 5′ in-frame kinase fusions in PDX models (Supplementary Fig. 3a). Despite a difference in distributions of 5′ and 3′ kinases, we observed higher combined percentages of 5′ kinases and "both-kinases", i.e. both 5′ and 3′ are kinases, in most cancer types (Fig. 4a). This observation accords with the hypothesis that 5′-kinases are more likely to be functional, since the promoter and other upstream regulation complexes are intact. To further evaluate the expression statuses of in-frame kinases, we compared kinase fusions shared between TCGA and our cohort. Most of the samples with in-frame fusion kinases have higher kinase expression compared to those without fusions in both cohorts (Fig. 4b).

We also assessed the overall landscape of gene expression involving fusion events across all PDX samples (Supplementary Fig. 3b), focusing on those oncogenes and tumor suppressors identified by TCGA in specific cancer driver contexts[40]. Fusion events involving oncogenes are uniformly upregulated compared to wildtype samples. For instance, EGFR is strongly up-regulated in BLCA, HNSC, and STAD PDX models, and ERBB2 in HNSC, making them good candidates for treatment studies. To provide deeper insight on the effects of fusion events on downstream pathways, 2 fusions, SS18-SSX1 in SARC and FGFR3-TACC3 in HNSC, and their effects on downstream pathways are illustrated in detail. For dots in the violin plots (Fig. 4c, d), each color indicates samples from a given PDX model and the diagram below these panels depicts the regulatory mechanisms with potential routes for treatment intervention. SS18-SSX is an important oncogenic event in synovial sarcoma[41]. SS18, SSX1, and another SWI/SNF complex member TLE1[42] are elevated in 3 PDX models (16 samples) by SS18-SSX1 fusions in SARC, as key target genes AXIN2, MYC, and CCND1 in the Wnt pathway[43] and IGFB2 and IGF2[44] for IGF driven tumor genesis (Fig. 4c, right panel). Histone deacetylase (HDAC) inhibitors have shown promising tumor suppressing effects in synovial sarcomas with SS18-SSX both in vitro[45,46] and in vivo[46]. In addition, the HDAC inhibitor quisinostat rescues early growth response 1 (EGR1) and CDKN2A expression by disrupting the SS18-SSX driven protein complex[47], while the former was also found to be rescued by the HDAC inhibitor romidepsin in vitro[45]. Both genes were found to be suppressed in most SS18-SSX1 fusion PDX models (Fig. 4c, right panel) and therefore could also be downstream reporters of the treatment efficacy.

Another well-known fusion, FGFR3-TACC3, activates the RAS, MAPK, and PI3K pathways[48] and is upregulated in HNSC PDX samples (Fig. 4d). We also observed upregulation of several key genes in FGFR downstream pathways, including RAS-MAPK, JAK/STAT, and PI3K-ATK, which promote cellular proliferation, migration, angiogenesis, and anti-apoptosis[49] and could therefore confer survival advantages for affected cells. PIN4, an intermediate of the FGFR3 downstream pathway to mitochondrial metabolism[50], is similarly affected. These observations point to FGFR inhibitors, such as

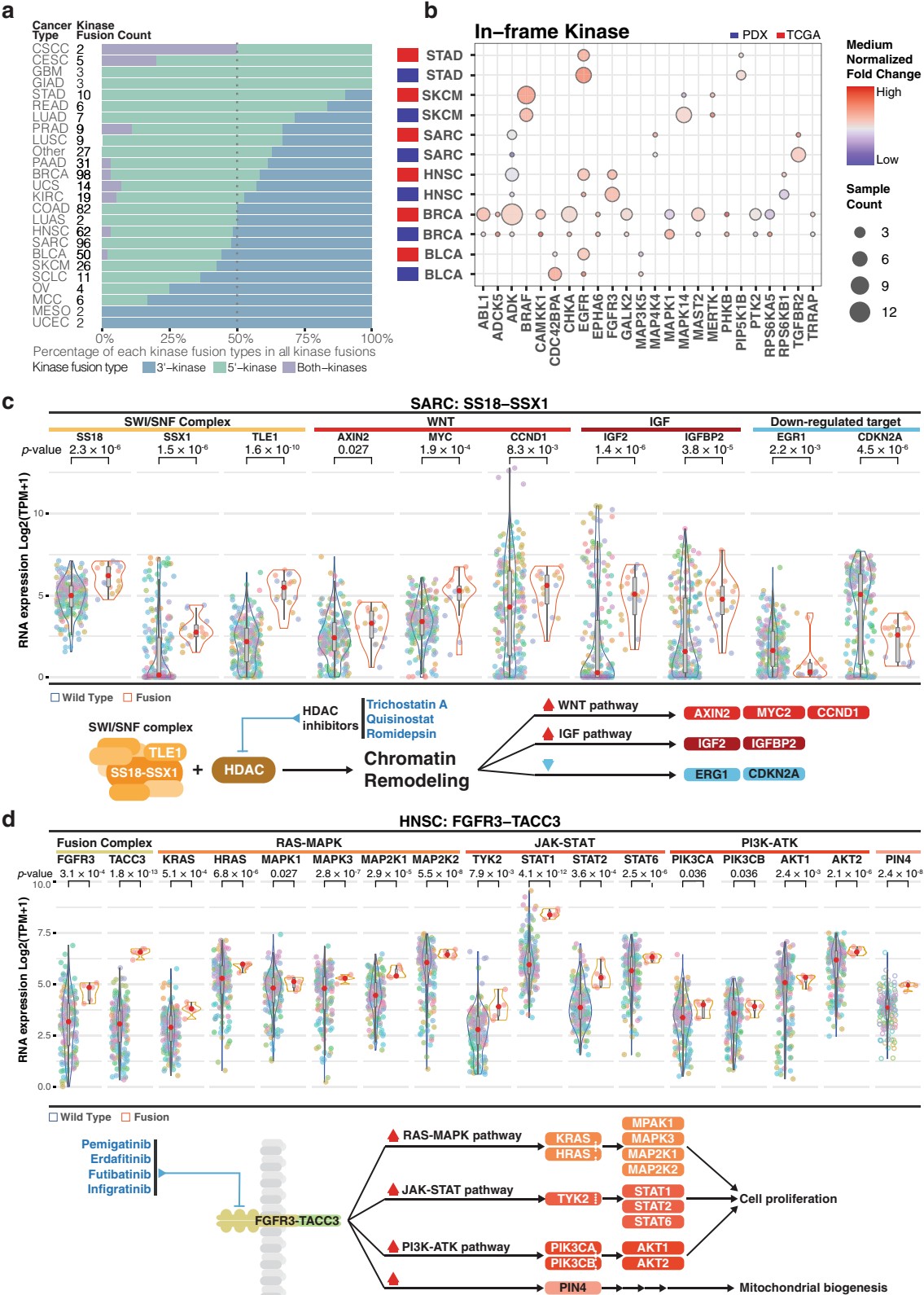

pemigatinib, futibatinib, and infiigratinib for testing treatment efficacy. These HNSC PDX models are also eligible for NCI-MATCH trial arm K2 testing the FGFR inhibitor erdafitnib. These examples showcase PDX models suitable for further treatment studies and how downstream pathways could be used as reporters for evaluating treatment efficacy.

**Pan-cancer transcriptional groups in PDX.** Similar molecular features can characterize more than one cancer type, suggesting the use of PDX models to extend drugs beyond current approved single cancer type treatments and the possibility of grouping to increase statistical power. We classified PDX models according to the top 1000 most variable genes from cancer types with more

**Fig. 4 Fusion events in pan-cancer. a** Distribution of kinase fusion type based on kinase location (5′, 3′ and both kinases fusion) per cancer type (Fusion count: kinase fusion events detected in each PDX model. Kinase fusion percentage: kinase fusion count/total detected fusions). **b** Median normalized expression of fusion involving kinase per cancer type. Fusion in greater than 2 PDX samples while overlap with TCGA events are shown (Sample count: independent PDX sample). **c, d** Expression of genes in fusions and those significantly altered (Wilcoxon test, p < 0.05) in the downstream pathways for respectively (**c**) SS18-SSX1 in sarcoma (SARC) and (**d**) FGFR3-TACC3 in head and neck squamous cell carcinoma (HNSC). The diagram below illustrates the simplified mechanism and pathway for given fusion. Potential treatment intervention labeled in blue texts. Dot color indicates samples from the same patient case. The diagram below illustrates the simplified mechanism and pathway for given fusion. Potential treatment intervention labeled in blue texts. The box boundary of each box plot indicates third quartile and first quartile respectively from the top to bottom. The whisker on top were drawn out from the third quartile to the largest data point or up to 1.5 × IQR. Similarly, the bottom whisker extends from the first quartile down to 1.5 × IQR or the lowest data point. The red dot at the center indicates medium. P-values were calculated using two-sided Wilcoxon rank-sum tests. Source data are provided as a Source Data file.

than 20 samples using ConsensusClusterPlus[51] and showcase the positive significant differentially expressed genes (FDR > 0.05, fold change > 1) from each transcriptional group (Fig. 5a). From the pan-cancer clustering analysis, we identified 4 major transcriptional groups that cluster cancer types according to cell-of-origin or organ system. This finding is consistent with TCGA results[52]. Here, groups 1 through 3 are enriched respectively with squamous cancer types, BLCA, HNSC, and LUSC, cancers of connective tissues, SARC and SKCM, and digestive system cancers, COAD, READ, and PAAD (Fig. 5a, b). Group 4 is a mixture of types (Fig. 5a, bottom right pie chart) having relatively low gene expressions overall. Very few positive differentially expressed genes (DEGs) were found for these cases (Fig. 5a) and it locates essentially at the intersection of groups 1 through 3 (Fig. 5b, system panel). Clustering depends less on PDX passage, treatment status, or racial group, and rather more on cancer types with similar organ systems (Fig. 5b, cancer group and system panel, and Supplementary Fig. 5), as described above.

We also analyzed driver oncogenic pathways, finding that Wnt is enriched in pan-cancer transcriptional group 3 (Fig. 5c). This observation accords with previous findings[53] since group 3 is enriched with COAD and roughly 80% of COAD cases are driven by Wnt activation. Another important aspect is the similarity of the expression profiles from the same PDX model. Here, we defined the cluster shift score as the percentage of PDX samples that have different cluster assignments from the same PDX model and use it as a metric for PDX similarity. Higher score indicates higher similarity. The majority of PDX models have high score close or equal to 1 (Fig. 5d, first panel), indicating that gene expression profiles remain consistent for most models. These models tend to cluster closely in UMAP plots (Fig. 5d, right panel). However, a few PDX models have samples with higher expression diversity, one example being PDMR-521955 having 4 different PDX models: R2, R3, R4, and R6 (Fig. 5d second and the third panel). All 4 models are metastatic PAAD tumors from a Caucasian female in her early 60 s. Each model comprises multiple passages of PDX samples from different passing paths (Fig. 5d second panel). Collectively, these 4 models formed 2 major UMAP clusters with 1 sample (G26) deviating the most. We found that samples with lower overall expression level move toward transcriptional group 4 on the UMAP. Indeed, the G26 outlier has the lowest overall gene expression profile of all samples from case PDMR-521955 (Supplementary Fig. 4). This observation implies some PDX models experience transcriptional alterations resulting in an overall lower expression level that deviates from the original tumor sample. Upon further investigation, we did not identify any statistically significant correlations of cluster shift scores with either tumor purity (Wilcoxon test, p > 0.5), mutation count (Wilcoxon test, p > 0.1), or any specific mutations. Therefore, identification of cluster shifts at the pan-cancer level might guide the selection of PDX model and aid in interpreting further treatment testing results.

Overall, using gene expression from PDX and human samples for clustering, we were able to recapitulate the coalesce of cancer types with similar organ systems (transcriptional group 2 and 3) or cell-of-origin (transcriptional group 1), while identifying a group of samples with overall low gene expression (transcriptional group 4). These clustering results could help elucidate cancer types with similar pathway activations and guide selection of PDX samples for treatment testing according to their similarity in gene expression profiles.

**Whole-genome duplications (WGDs) in multiple PDX passages and subclonal evolution.** CNAs can independently alter copy numbers of each of the two alleles, resulting in different somatic events whose identification requires allele-specific information[54]. An example is WGD, which is frequent in cancer and associated with poor prognosis[55,56]. We used HATCHet[57] to search for allele-specific CNAs and WGDs in 270 PDX samples from 54 cases having available matched-normal samples. HATCHet jointly analyzes changes in read-depth and germline SNP frequency across samples from the same case to identify allele-specific CNAs (Fig. 6a). Its results quantified levels of WGDs and loss-of-heterozygosity (LOH) events and characterized tumor clonal compositions (Fig. 6b).

We found WGDs in 128 human and PDX samples from 27 cases, with a frequency of 50%. WGD presence is well supported by sequencing data in each sample by clusters of genomic regions with distinct values of read depth and levels of allelic imbalance, which are absent in samples without WGD (Fig. 6c). Such clusters are a hallmark of WGDs[57], but we further correlated WGDs with two other somatic events associated with WGDs[56,58], namely accumulation of deleterious events and LOH of TP53. WGD was correlated with abundance of deletions ($p = 2.85 \times 10^{-23}$, chi-square test), the latter determined by counting samples in which the fraction of the genome affected by deletions was higher than that affected by amplifications. Results are consistent with the findings of Lopez et al.[58] for non-small-cell lung cancer patients, in which it was suggested that such correlation is explained by a selective pressure for WGDs to mitigate the effects of deleterious alterations.

TP53 LOH events prevent genome-doubled cells from re-entering the cell cycle and proliferating[59]. We found these to be correlated with WGDs ($p = 1.39 \times 10^{-10}$, chi-square test) and strongly supported by sequencing data, since most samples with a WGD exhibit clear shifts of allelic frequencies for germline SNPs genomically close to TP53 in chr17 indicating the presence of a single allele. In contrast, we observed several samples, both with and without such shifts, across samples without a WGD (Fig. 6d). Notably, we also found that abundance of deletions and TP53 are significantly correlated ($p = 1.73 \times 10^{-8}$, chi-square test), suggesting an important interplay among these phenomena.

Finally, we investigated clonal structure using CNAs inferred by HATCHet in distinct subpopulations of cells, finding subclonal CNAs present in 96 samples. We also found significant correlation

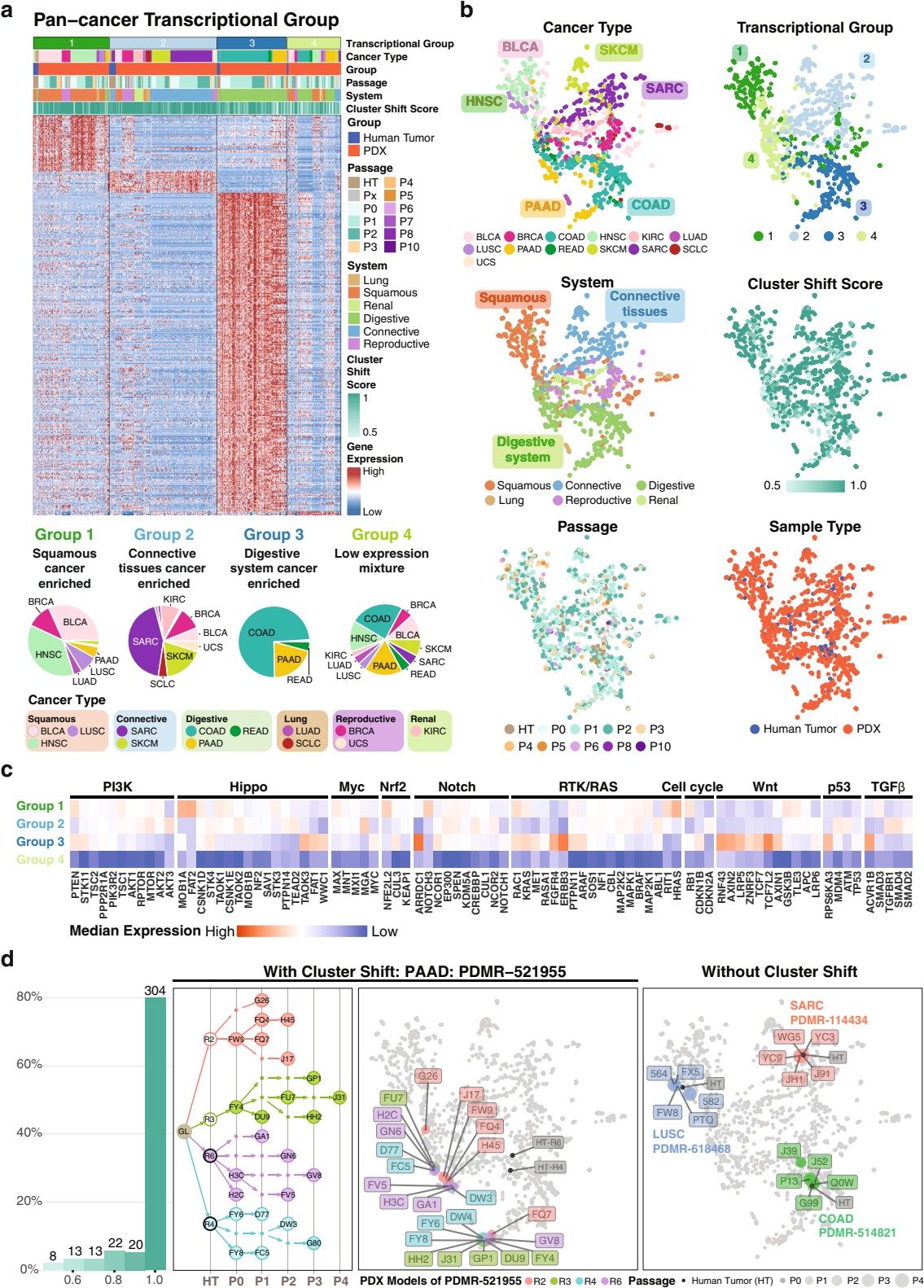

($p = 2.05 \times 10^{-7}$, chi-square test) between subclonal CNAs and LOH events in *TP53*, supporting the view that *TP53* mutations lead to higher genomic instability. Using HATCHet, we also searched for samples with CNAs that are not present in other samples from the same case (inter-sample subclonality), finding 75 instances. Subclonal CNAs and different CNAs across samples from the same case suggest ongoing clonal dynamics between multiple samples

from the same PDX. For example, we found different tumor clones for colon cancer case PDMR-519858 between samples from the primary tumor and different PDX passages (Fig. 6e).

**Prediction of candidate PDX models by NCI-MATCH treatment arms.** We compared variants identified in human and PDX

**Fig. 5 Pan-cancer transcriptional groups. a** Top panel is four major pan-cancer transcriptional groups (limegreen, skyblue, oceanblue, and light olive) identified from the expression of significant positive differentially expressed genes (FDR > 0.05, fold change > 1) of each transcriptional group in cancer types with sample size greater than 20. Those include Bladder Urothelial Carcinoma (BLCA), Breast invasive carcinoma (BRCA), Colon adenocarcinoma (COAD), Head and Neck squamous cell carcinoma (HNSC), Kidney renal clear cell carcinoma (KIRC), Lung adenocarcinoma (LUAD), Lung squamous cell carcinoma (LUSC), Pancreatic adenocarcinoma (PAAD), Rectum adenocarcinoma (READ), Sarcoma (SARC), Small cell lung cancer (SCLC), Skin Cutaneous Melanoma (SKCM), Uterine Carcinosarcoma (UCS). The heatmap shows differentially expressed genes (DEGs) from each group. Bottom panel is the ratio of each cancer type in each transcriptional group. **b** Dimension reduction UMAP 2D-plots using 1000 most variable genes. Each point represents one sample. Colors in each panel indicate respectively cancer type, transcriptional group, system, cluster shift score, passage, and sample type of each sample. **c** Normalized medium expression of genes in major oncogenic pathways. **d** First panel is the distribution of cluster shift score. Second and third panel are the pedigree tree and the 2D distribution of case PMDR-521955. Each color (light coral, yellow green, purple, and light teal) indicates one PDX model originated from the same hurman tumor sample. In the second panel, filled circles are human samples with RNA-Seq data. Hollow circle and dot are human and PDX sample without data. Fourth panel is an example of PDX models without cluster shift. Each color (red, blue, and green) indicates samples from one PDX model. Source data are provided as a Source Data file.

tumors with those being studied in the National Cancer Institute Molecular Analysis for Therapy Choice (NCI MATCH) program[9,10], a phase II clinical trial that seeks to determine treatment effectiveness based on genomic alterations, regardless of the cancer type. We first discerned somatic alterations in PDX models that satisfied study arm specifications and then applied disease exclusion conditions (we also considered clinical biomarkers such as HER2 status). Among the 38 study arms, we found 25 target genes that have non-silent mutations across 22 cancer types (Fig. 7a). We also found 22 recurrent mutations across 13 cancer types (>1 PDX model) in 10 druggable target genes: *PIK3CA*, *FGFR* (*FGFR1*, *FGFR3*), *AKT1*, *BRAF*, *PTEN*, *BRCA2*, *ERBB2*, *KIT*, and *NF1*, which, respectively, matched NCI-MATCH drugs copanlisib, erdafitinib, capivasertib, dabrafenib with trametinib (or ulixertinib), GSK2636771, adavosertib, afatinib, sunitinib malate, and trametinib (Fig. 7b).

*PIK3CA* was enriched with non-silent mutations in BRCA and COAD PDX models (Fig. 7a) and is the most commonly mutated gene, with 8 recurrent point mutations (Fig. 7b). In particular, E545K appeared in 16 PDX models and was frequently detected in COAD PDX models. In addition, H1047R and E542K hotspot mutations frequently occurred in BRCA (Fig. 7b). These three mutations, E542K, E545K, and H1047R, are PIK3CA hotspot members[60]. H1047R is associated with a lower pathological complete response rate in triple-negative BRCA patients treated with anthracycline-taxane-based neoadjuvant chemotherapy[61]. Regarding structural variations, BRCA-derived PDX models had clear, frequent amplifications and deletions in *FGFR1* and *PTEN*, respectively, as compared to other cancer types. Furthermore, *FGFR1*, *CCND1*, and *PIK3CA* genes are frequently amplified in the HNSC- and LUSC-derived PDX models, and *PTEN* is frequently deleted in BRCA-derived PDX models (Fig. 7a). As expected, these patterns are consistent with previous observations[62,63]. There were also several fusion PDX models observed across cancer types (Fig. 7a). Overall, 258 unique PDX models across 23 cancer types and the mixed cancer type (Other) were identified as potential candidates for clinical trials (Fig. 7a and Supplementary Data 3).

We then sought to determine the number of relevant study arms for every PDX model. From 258 candidate models (Fig. 7c), approximately 62% matched a single study arm ("single-arm" event) while 38% ($n = 98$) matched multiple arms. In addition, among the 897 PDX samples generated from these 258 PDX model lines, around 83% matched the current target arms with over 200 distinct genetic alterations. Databases such as DEPO[64] and CIViC[65], where druggable alterations are generally associated with specific cancers rather than with pan-cancer, reveal that nearly 35% of these alterations are reported with high confidence, leaving up to about 65% as potentially novel. A further comparison reveals that 76% ($n = 120$) of this 65% are listed in either of the TCGA or COSMIC (v90) human cancer databases, leaving 24% ($n = 38$) as yet uncharacterized (Fig. 7c, Supplementary Fig. 6a, and Supplementary Data 4). Additional drug databases would be expected to inform these percentages further.

To identify PDX samples that match well with arm targets for drug trials, we searched for somatic alterations associated with gene expression across druggable target genes (Wilcoxon test, $p < 0.05$), finding 30 such alterations among 18 genes and 15 cancer types that are significantly different from their wild types (Supplementary Data 5). The mutation events were found across 14 mutated genes and were associated with gene expression level changes. The three most statistically significant oncogene mutations among diverse arms were FGFR3 S249C, BRAF V600E, and PIK3CA D1017N in BLCA, PAAD, and STAD, respectively. These alterations matched, respectively, with the drugs erdafitinib, dabrafenib with trametinib, and copanlisib (Fig. 7d). Fusion events were detected in *BRAF* and *FGFR3* in SKCM and HNSC cancer types, respectively, with high expression as compared with wild types. These alterations matched with trametinib and erdafitinib, respectively (Supplementary Data 5). Copy number amplification events in *CCND1*, *CCND3*, *CDK6*, *ERBB2*, *FGFR1*, *MET*, and *PIK3CA* also showed higher expression compared to wild types across various cancer types. These alterations matched with palbociclib, trastuzumab with pertuzumab, erdafitinib, and crizotinib (Fig. 7e). These PDX models, with their drug-targeting recurrent alterations and gene expression validation, are strong candidates for in vivo drug tests. Interestingly, some alteration arms have already been supported by recent literature of clinical trial, cell line and PDX study (Supplementary Data 5).

## Discussion

TCGA studies examined driver events in various human cancers and provided druggable candidates for clinical drug trials[40,66,67]. PDXs have been designed as in vivo models for studying drug response by virtue of capturing the principal genomic features of human tumors[6,68]. Understanding the true nature of PDX genomic features through quantification of their similarity to human tumors is vital for studying drug response trials using PDXs. Here, we characterized the genomic features of 536 PDX models and 268 human parental tumors across 25 cancer types.

By comparison to primary human tumors, variant allele fractions (VAFs) in PDXs were found to be higher overall, which reflects high tumor purity and the selection of sub-tumor clones in PDXs. For instance, VAFs of TP53 hotspots (R175H, R248Q, R248W, and R273H) were close to 1 in PDXs, while being close to 0.5 in human primary tumors, suggesting a loss of heterozygosity (LOH) in PDX. We generally found high concordance of key driver mutations in PDXs and their corresponding human tumors, though a small fraction ~10% were discordant, suggesting

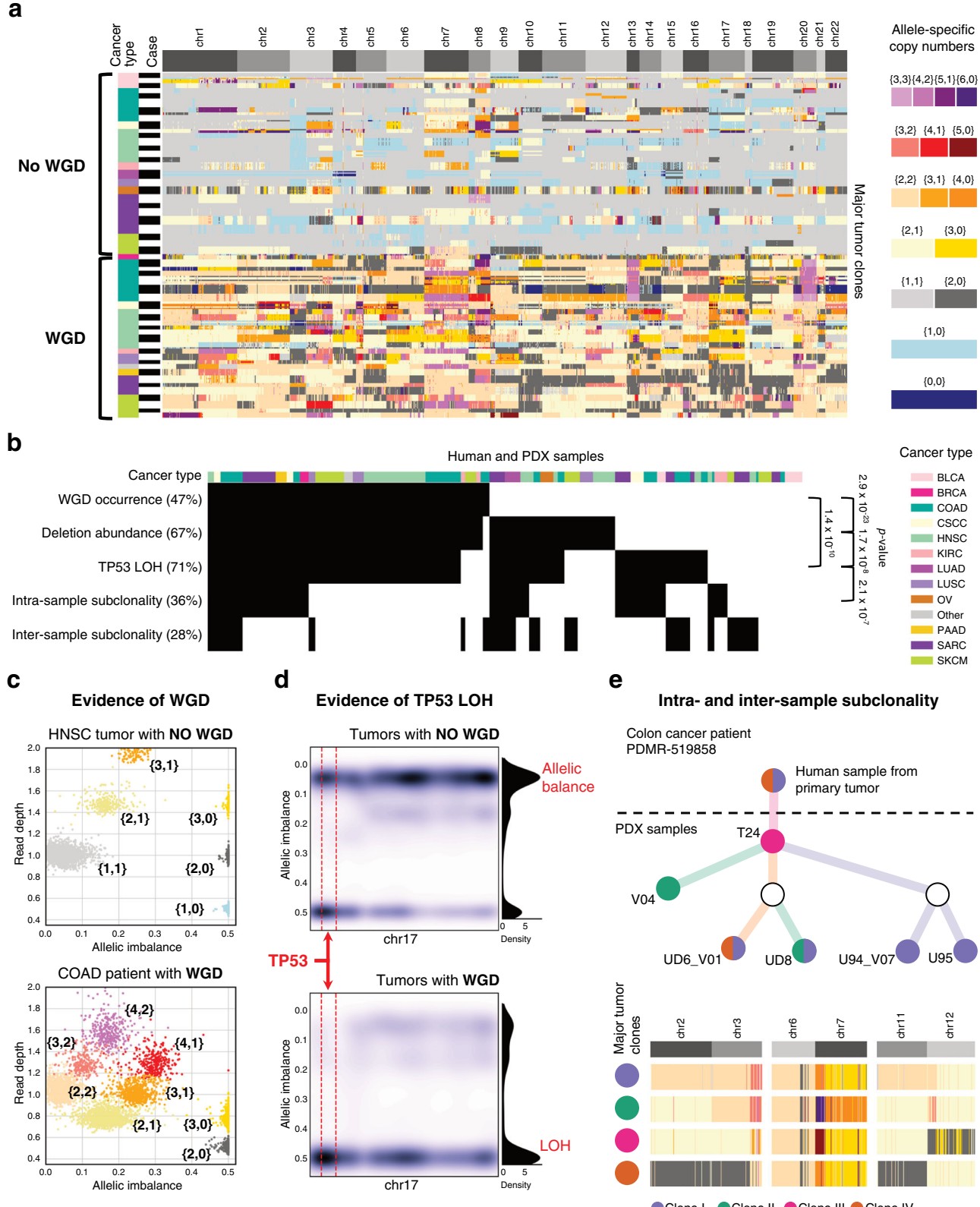

the existence of tumor clonal evolution. Analysis of cases with multiple PDX models shows a higher intra-mutational similarity compared to inter-mutational similarity, suggesting intrinsic tumor heterogeneity among different tumor segments. In contrast to these mutational features, we observed a rapid CNV evolution in several PDX models, consistent with a previous study[69].

In accordance with recent pan-cancer studies[55,56], we observed that whole-genome duplications (WGDs) are relatively frequent (~50%) across human and PDX samples. Notably, we also confirmed a significant correlation between WGDs and both abundance of deletions and LOH of TP53[56,58]. Moreover, our study shows that the LOH of TP53 is also significantly correlated with

**Fig. 6 Extensive presence of WGDs and subclonality correlates with abundance of deletions and TP53 LOH. a** Allele-specific CNAs of major tumor clones and presence/absence of WGD are inferred for 270 human and PDX samples from 54 cases of different cancer types. **b** Presence of WGD, deletion abundance, LOH of gene *TP53*, intra- and inter-sample subclonality are indicated (black) across all samples and significant correlations between pairs of these features are reported on the right side (*p*-values are computed from Pearson's chi-squared statistic). **c** The absence/presence of a WGD (top/ bottom) are supported by the presence of low/high numbers of distinct clusters of genomic regions (each point corresponds to a 250 kb genomic bin and is colored according to the corresponding allele-specific CNAs using the same color legend as in **a**) with different values of allelic balance and read depth in two PDX samples from two HNSC and COAD patients. **d** A kernel density estimate of the allelic balance is computed for all samples without (top) or with (bottom) a WGD across 250 kb bins of chromosome 17 (whole-chromosome density is shown on the right side) and in a 6 Mb genomic region surrounding gene *TP53* (dashed red lines). **e** For a colon cancer (COAD) patient, a tree represents the relationships between the corresponding human and PDX samples (nodes), which contain four major tumor clones (violet, green, magenta, and dark orange) with different CNAs (bottom with allele-specific CNAs colored as in **a**). Source data are provided as a Source Data file.

the presence of multiple tumor clones in the same sample (intra-sample subclonality).

Strong cis and trans mutational effects of several key driver genes were identified in PDXs, in accordance with human tumors from TCGA studies. Specifically, we found down-regulated expressions in PDX samples harboring mutations in tumor suppressor genes, such as *APC*, *RB1*, *KMD6*, and *STAG2*[53]. For oncogenes, such as *BRAF*, *FGFR3*, *HRAS*, and *KRAS*, we observed high expression in mutated samples, which suggests these mutations may be activation events. Interestingly, we also found in both PDX and TCGA human tumors with mutations in melanoma (SKCM) a high expression of *CDKN2A*, which has not typically been classified as an oncogene. In terms of trans effect, we found high *MYC* and *PLK2* expression in *APC* and *FBXW7* mutated PDX samples in COADREAD, respectively. Notably, high *RHEB* expression was observed in *BRAF*-mutated PDXs in SKCM, suggesting a target for PDX drug response trials beyond the known BRAF and MEK inhibitors. Also, we observed the activation of the NRE2 pathway in *NFE2L2*-mutated PDX samples in squamous cell lung cancer (LUSC), suggesting that PDXs may be suitable for studying key oncogenic pathways and corresponding drug responses. In addition, we identified many PDX models carrying CNV amplification events in *CCND1/3*, *CDK6*, *ERBB2*, *FGFR1*, *MET*, and *PIK3CA*, which correspond to NCI-MATCH clinical trial drugs (palbociclib, trastuzumab with pertuzumab, erdafitinib, AZD4547, crizotinib, and taselisib). For instance, *CCND1* amplifications were observed in sarcoma (SARC) PDXs, which match the trial drug palbociclib. In addition, we identified four distinct pan-cancer groups in the current PDX cohorts representing the different origins for these tumors, namely squamous cells, connective tissues, the digestive system, and a mixture. The Wnt oncogenic pathway is upregulated in the digestive system group, suggesting intriguing group-specific targets for PDX drug clinical trials.

Although our study is primarily computational, independent pharmacological experiments involving the same PDX models used here have indirectly validated some of our drug target results. For instance, PDX model WM3936-1 having BRAF hot-spot mutation p.V600E shows good response to BET and MEK inhibitors[29]. Drugs targeting the PI3K pathway show promising clinical response in PDX models HCI-003, HCI-013, WHIM12, and WHIM20 having PIK3CA mutations[32–34]. Finally, recent treatment studies show good response for GLUT inhibitors in lung cancer PDX models TC333, TC453, and TC494 carrying *KEAP1* or *NFE2L2* mutations[37].

In summary, the present study represents the largest-scale comprehensive genomic characterization of PDX models, including driver mutations, fusions, and CNVs. The observed identities and differences between PDX genomic features and their corresponding human primary tumors will be an important resource for future PDX studies. The key driver events we observed and the corresponding cis and trans effects on gene expression provide therapeutic targets for future PDX drug response trials. A limitation of the current study is the relatively low PDX model counts for certain cancer types, including GBM, KIRC, STAD, and OV, despite the pooled contributions of our consortium members, each with its own cancer type specialties, which can depend on geographical location, medical and scientific expertise, and other factors. Numbers are also a reflection of these cancer types' relatively lower incidence rates (seer.cancer. gov/statfacts), at 60 to 260 new cases per million annually, as compared to other common cancer types, such as breast and certain lung cancers with over 1,000 new cases per million annually. The PDX models for these low-incidence cancers are therefore all the more valuable. As the PDX community continues to grow and engage with cancer centers worldwide, the representation of cancer types will become wider and deeper, allowing for the identification of more druggable events and the discovery of further insights into PDXs' recapitulation of human tumors.

## Methods

**Sample collection and dataset**. Experimental details for PDX model sources appear in Supplementary Methods. Sequence files consisted typically of patient tumor/normal matched samples and of PDX samples from one or multiple passages. The available whole-exome sequence (WES) and RNA-sequence (RNA-seq) data for human and PDX samples were downloaded from various sources (see Data availability) for local processing. Various QC metrics, including overall coverage and lineage consistency (see below) were computed and used to validate the samples. To support pan-cancer analysis, diagnoses from the Cancer Therapy Evaluation Program (CTEP) were mapped to The Cancer Genome Atlas (TCGA) study codes or designated as cancer type "other." The resulting sample set consists of 511 patient cases across 25 cancer types and 536 PDX models. These selected cases usually consist of one or more PDX passages and a variable number of human samples, the availability of which determines the particular variant calling pipeline to use. The clinical data included patient demographics, patient tumor status (i.e., primary or metastasis), and whether patients received any treatment prior to PDX collection.

**Precision medicine NCI-MATCH trials**. Genetic eligibility criteria were compiled from resources at the NIH NCI-MATCH trial website (www.cancer.gov/about-cancer/treatment/clinical-trials/nci-supported/nci-match; accessed 29 April 2020), the ECOG-ACRIN Cancer Research Group website (ecog-acrin.org/trials/ nci-match-eay131, along with downloadable Excel spreadsheet versioned 26 April 2020; accessed 29 April 2020), and the NIH clinical trials website (ClinicalTrials. gov identifier NCT02465060; accessed 1 May 2020). Disease exclusions were taken into account in reporting our results.

**Raw reads filtering and mouse reads filtering**. All WES and RNA-Seq data underwent initial processing to trim adaptors and filter poor quality reads using Trim Galore (v0.5.0) (www.bioinformatics.babraham.ac.uk/projects/trim_galore). For PDX model sequence data, Disambiguate (v1.0)[70] was used to filter mouse-derived reads in WES and RNA-Seq data using mouse (GRCm38, GENCODE vM19, https://www. gencodegenes.org/mouse/releases.html) and human (GRCh38, GENCODE v29, https://www.gencodegenes.org/human/releases.html) reference genomes. The resulting WES reads were then deduplicated and converted to bam format using Samtools (v1.5, https://www.htslib.org), Picard (v2.20.1, https://broadinstitute.github.io/picard), and BWA-MEM (v0.7.17, https://github.com/lh3/bwa) for use in downstream analysis.

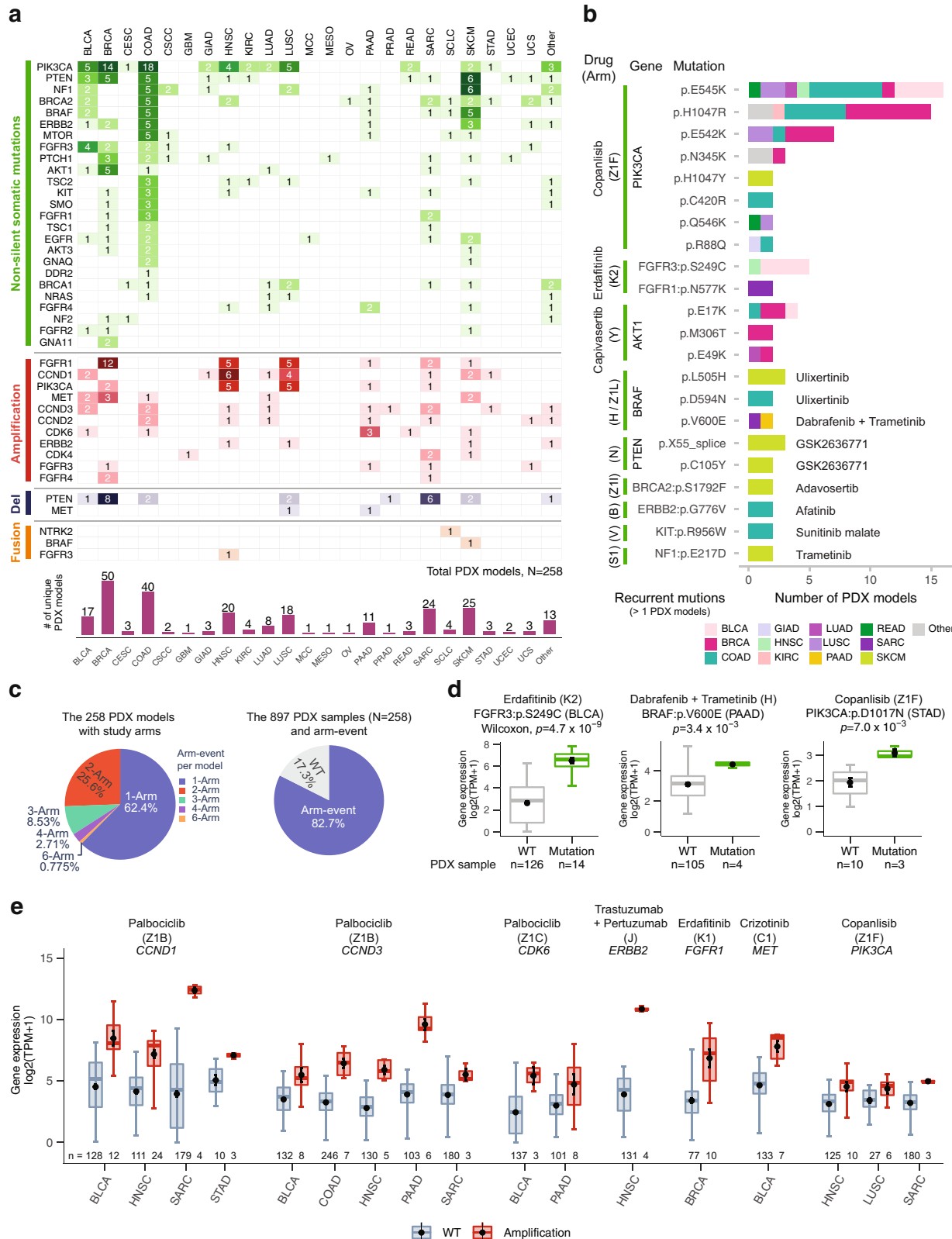

**Sample quality estimate**. To check for consistency across sample lineages, we developed a sequence data quality control algorithm named SeqQEst (Beta version, github.com/ding-lab/SeqQEst) to detect potentially mislabeled, swapped, and tissue-contaminated samples that occasionally appear, especially in large cohorts of sequencing data. SeqQEst has two main pipelines: (1) Sequence QC (SeqQC): to estimate bam file status (total reads, mapping percentage, average read length, mean mapping quality, insert size, and coding region coverage) and to provide a FastQC summary report; (2) Sample germline QC (GermlineQC): to detect sample

swap/mismatch/contamination across a collection of samples by correlating variant allele frequencies of a target set of germline SNPs. The target SNP markers were selected from ~5 million common missense SNPs (dbNSFP v3.5a; https://sites. google.com/site/jpopgen/dbNSFP and Pengelly et al., 2013[71]) across chromosomes 1–22 in the human genome based on WES and RNA-Seq data testing. Filtering out contaminated or swapped samples using GermlineQC resulted in good agreement with short tandem repeat (STR) polymorphism data. For samples remaining ambiguous after GermlineQC, we applied HLA-QC. HLA-QC uses major

**Fig. 7 NCI-MATCH trial related druggable genes and recurrent alterations. a** Genomic alterations of 258 candidate PDX models, which satisfied the study arm specifications across 23 cancer types, include the 25 target genes of non-silent mutations, 13 copy number alteration genes, and 3 fusion genes. **b** Distribution of 22 recurrent point mutations (> 1 PDX models) in the 10 druggable target genes with 9 drugs across 12 cancer types. The information in parentheses is a matched study arm. **c** Distribution of target arms per PDX model and positive signal arms in PDX samples. The left pie chart presents the distribution of PDX models in single-arm and multiple-arm (>1 arms). The right chart shows the percentage of PDX models between wild type and arm-event. **d, e** Target drugs and target gene alterations that are associated with gene expression (*N*, number of independent PDX models; *n*, number of independent PDX samples). **d** The point mutations of target arms that relate to upregulation of gene expression in cancer. **e** The amplification-related target arms match with upregulation of gene expression in different cancer types (*n*, number of PDX samples). And gene expressions are significantly different (absolute value of log2-fold-change > 0.585 and *p* < 0.05) between wild-type (WT) and amplification. In **d** and **e** data are presented as box plots where the middle line is the median, the largest value at the end of the upper whisker is the maximum, and the smallest value at the end of the lower whisker is the minimum. The black dot with error bar is mean ± SEM in box plots. The *p*-values are calculated by a two-sided Wilcoxon rank sum test in R. Source data are provided as a Source Data file.

---

histocompatibility complex (MHC) class Ia loci (HLA-A, -B, and -C) to assess data contamination or swap. Samples passing the GermlineQC analysis and having sufficient coverage (> 20x coding region coverage in WES or > 25 Mb mapped depth in RNA-Seq data) were passed to downstream analysis.

**Somatic mutation calling**. Somatic mutations were determined using our in-house pipeline SomaticWrapper (v1.5, github.com/ding-lab/somaticwrapper), which is anchored by four somatic variant calling tools: Strelka (v2.9.2)[72], Mutect (v1.1.7)[73], VarScan (v2.3.8)[74], and Pindel (v0.2.5)[75]. For candidate somatic mutations, low quality instances were filtered by bam-readcount (github.com/genome/bam-read-count) using parameters –q 10 –b 20. To generate high confidence mutation calls, we only kept the mutations that were supported by at least 2 callers and satisfied cutoffs of at least 14 total reads in the tumor and at least 8 in the normal. The mutations were further filtered by discarding observed variant alleles in fewer than 4 reads and those having variant allele fractions (VAF) less than 0.05 in tumor or higher than 0.01 in normal.

**Tumor-only somatic mutation calling**. Tumor-only somatic variants were called using Mutect2 (v4.1.2.0) best-practice pipeline (In-house scripts: github.com/ding-lab/PDX-PanCanAtlas/tree/master/data_process/somatic.Mutect2_tumorOnly) with the GDC Panel of Normal (PON) data (gdc.cancer.gov/about-data/gdc-data-processing/gdc-reference-files; gatk4_mutect2_4136_pon.vcf.tar). To reduce false positives further, we used only those mutation sites having ≥ 20× coverage and > 3 reads supporting mutations with ≥ 0.1 tumor VAF, which were supported by bam-readcount evidence.

**Extra false-positives filtering in somatic mutations**. Potential false-positive calls can arise from sequencing or alignment errors in low mappability regions[76]. In general, PDX samples have higher false-positive mutation sites than human samples due to mouse homologous reads, even after removing contaminating mouse reads. To increase the overall confidence level of human somatic variant calls, we applied the following steps. Somatic mutations of PDX samples that were retained if they were reported in COSMIC (v90, https://cancer.sanger.ac.uk/cosmic) or TCGA Cohort (https://gdc.cancer.gov/about-data/publications/mc3-2017). Furthermore, calls in PDXs that had a matched human tumor were retained if the variant was present in both the PDX sample and in the human tumor, regardless of whether it was in the COSMIC database. Finally, we removed point mutations located near indel regions (window size, 20 bp).

**Germline mutations calling**. Germline mutations were determined using our in-house pipeline GermlineWrapper (v1.1, github.com/ding-lab/germlinewrapper), which applies several germline variant calling tools, including GATK (gatk.broadinstitute.org), VarScan (v2.3.8)[74], and Pindel (v0.2.5)[75]. To generate high confidence mutation callings, we used the SNPs and INDELs supported by both VarScan and GATK, as well as INDELs reported by Pindel.

Variants called were filtered based on coding regions of full-length transcripts from Ensembl release 95 plus additional two base pairs bordering each exon in order to cover splice sites. We also required variants to have Allelic Depth (AD) ≥ 5 for the alternative allele. After filters, a total of 7,331,296 variants (~24,851 per sample) and 5,350,478 variants (~23,262 per sample) were kept for cases with matched tumor-normal samples and tumor-only samples, respectively.

The quality of variants passing all filters was assessed by calculating concordance with dbSNP (release 151, https://ftp.ncbi.nih.gov/snp/organisms/human_9606_b151_GRCh38p7/VCF) and the average transition-transversion (TiTv) ratio using GATK's VariantEval tool (v3.8 with default parameters). We obtained 98.95% overall concordance and 2.85 TiTv ratio for cases with matched tumor-normal samples and 95.44% overall concordance and 2.83 TiTv ratio for tumor-only samples.

**Pathogenicity assessment**. Annotation of germline variants that passed filters was performed using Ensembl Variant Effect Predictor (VEP) (v95 using default parameters, except where–everything)[77]. These variants were then assessed for pathogenicity using CharGer (v0.5.4)[78], which prioritizes germline variants according to published AMP-ACMG guidelines[79]. CharGer pulls information from ClinVar (release as of 08/15/2019 processed using github.com/macarthur-lab/clinvar), gnomAD (release 2.1.1, https://gnomad.broadinstitute.org), as well marshalling SIFT (v5.2.2)[80] and PolyPhen (v2.2.2)[81] in the implementation of 12 pathogenic and 4 benign modules for variant classification. We used the default CharGer scores for each evidence level (https://github.com/ding-lab/CharGer/tree/v0.5.4). The detailed implementation and parameters used here are at: https://github.com/ding-lab/PDX-PanCanAtlas/tree/master/analysis/CharGer. Variants were labeled, as follows: pathogenic if they were known pathogenic variants in ClinVar, likely pathogenic for CharGer score > 8, and prioritized VUS for CharGer score > 4.

Variants classified as Pathogenic or Likely Pathogenic were filtered for rare variants with ≤0.05% allele frequency in gnomAD (release 2.1.1). Cancer-relevant Pathogenic and Likely Pathogenic variants were selected based on whether they were found in the curated list of 152 cancer predisposition genes from Huang et al. (2018)[19]. Additionally, read count analysis using bam-readcount (v0.8 with parameters -q 10, -b 15) was performed in both normal and tumor samples in order to evaluate the number of reference and alternative alleles for each variant. Variants were required to have at least 5 counts of the alternative allele and a variant allele frequency (VAF) of at least 20%. Furthermore, variants common in our cohort (cohort MAF > 1%) were not considered.

Additional filtering steps were applied to variants from cases without matching normal samples available (i.e. tumor-only cases). First, we filtered all somatic mutations for each case from the list of obtained germline variants. Next, we filtered all variants that were present in the COSMIC database (v.86)[82]. Finally, we retained only variants present in the gnomAD database (release 2.1.1) that have a MAF ≤ 0.05% in order to concentrate our analysis on rare germline variants.

**Focal copy number alteration**. The somatic copy number alterations (CNAs) were predicted using CNVkit (v0.9.6)[83]. Matched tumor-normal samples, with the matched normal as reference determined CNAs of tumor. For tumor-only samples, we create a pooled reference from several blood normal samples that were collected from matched tumor-normal samples (≥50× mean coverage in coding region and average read length ≥ 100 bp). We then used this pooled normal reference to predict CNAs for tumor-only samples. Low-quality CNAs were filtered based on coverage (< 20), the number of probes (< 10), and length (< 5 kb). To define absolute copy numbers from CNVkit, the threshold is as follows: -t −1.3, −0.4, 0.3, 0.9. Deletion, loss, neutral, gain, and amplification of segment or gene-level defined as 0, 1, 2, 3, > 5 in absolute copy number. Gistic2.0[84] was used to predict chromosome arm copy numbers to compare to TCGA results. TCGA copy number results were downloaded from Firehose (gdac.broadinstitute.org).

**WGD prediction and clonal evolution**. We used HATCHet (v0.1)[57] to identify allele-specific CNAs and WGDs for multiple tumor clones jointly from multiple human and PDX samples from the same case. HATCHet requires three sources of information: a BAM file for each tumor sample, a BAM file for a matched-normal sample, and the reference genome used for the alignment of sequencing reads. Thus, we applied HATCHet to 270 human and PDX samples from 54 cases for which the required matched-normal sample was also available. Specifically, we applied HATCHet jointly on all samples from the same case and using the default values for all parameters, but increasing the minimum clone proportion to 10% due to the higher variability of whole-exome sequencing data than whole-genome sequencing data.

**Gene expression**. Kallisto (v0.44.0, default parameters)[85] was used to estimate transcript abundance with a GENCODE transcript reference (release 29, GRCh38).

We used the R package 'tximport' (v1.12.0)[86] to measure gene expression at the transcript level.

**Tumor purity prediction**. Tumor purity was assessed computationally in all paired samples using estimates derived from WES data and from RNA-seq data independently using ABSOLUTE (v1.0.6)[55] and ESTIMATE (v2.0)[87].

**Fusion**. For gene fusion detection, we used STAR-Fusion v.1.6.0 (github.com/STAR-Fusion), which identifies fusion transcripts from RNA-seq data and outputs all supporting data discovered during alignment. To remove false positive fusions, we used FusionInspector results that assists in fusion transcript discovery by performing a supervised analysis of fusion predictions, attempting to recover and re-score evidence for such predictions. To detect tumor fusions, we filtered non-cancer fusions via fusion annotation (e.g. GTEx_recurrent_StarF2019, BodyMap, DGD_PARALOGS, HGNC_GENEFAM, Greger_Normal, Babiceanu_Normal, ConjoinG), and previously reported normal fusions[88]. We further filtered fusions by FFPM (FFPM ≤ 0.1).

**MSI**. MSIsensor (v0.6)[89] and MSIsensor2 (v0.1, github.com/niu-lab/msisensor2) were used to distinguish microsatellite unstable (MSI) tumors from microsatellite stable (MSS) samples based on tumor/normal and tumor-only sequence data. The "msi" command was run with the default options and with the minimal homopolymer size set to 1 and minimal microsatellite size set to 1[40].

**Cis and Trans effect**. We examined *cis*- and *trans*-effects of significantly mutated genes (SMGs) of nine cancer types (BRCA, COADREAD, SARC, SKCM, PAAD, LUAD, HNSC, BLCA, and LUSC) based on previous large-scale TCGA pan-can study[14] on the RNA expression. After excluding silent mutations, samples were separated into mutated and WT groups. We used the Wilcoxon rank-sum test to report differentially expressed genes between the two groups and FDR correction is applied through standard R function "fdr". We use an FDR < 0.1 cutoffs for reporting differentially enriched genes. We further studied how the number of unique cis and trans events are affected by sample size by using BLCA as an example (Supplementary Fig. 2a). TCGA has more samples than our PDX cohort (406 vs 141) in BLCA. We performed the cis and trans analysis by subsampling TCGA data to 100, 200, 300, and 400 samples. Supplementary Fig. 2a shows the dependence of the number of unique cis and trans events from TCGA on sample size. We found the number of unique cis and trans events are highly correlated to sample size (Pearson's correlation R~0.999, *P*-value < 0.001), indicating that sample size has a major impact on the number of cis and trans events. However, when the sample size is close to the PDX sample, we still see unique events from TCGA, which may reflect the representation of different mutations in different sample sets even with the same sample size.

**Pan-cancer transcriptional grouping**. To highlight the most represented cancer types, those with sample sizes greater than 20 were selected for pan-cancer transcriptional grouping analysis. The expression data were first processed with the ComBat function in the R sva package (bioconductor.org/packages/release/bioc/html/sva.html) to remove batch effects between collection centers. The top 1000 most variable genes (defined by genes with the highest median absolute deviations) with less than 30% NA count were selected for unsupervised clustering using the Consensus Cluster Plus package[51]. Gene expressions were scaled across samples and clustered using default parameters for 1000 iterations. Optimum $k = 4$ value was determined using the elbow method and manual inspection of clusters with extremely low sample size. Differentially expressed genes for each cluster were used to plot the heatmap. Cluster shift score is a metric to measure the similarity of group assignment for PDX samples from the sample model. The score is defined by dividing sample count per group by the total sample count, then taking the maximum ratio as the score for that PDX model.

**Define a study arm match score for detecting positive signals**. To characterize the ability of PDX models to a satisfy study arm target criteria throughout their passages, we define a study arm match score, $S_{arm}$, as the fraction of unique passages across the cohort that displays a positive signal for the target. Here, the unique passages are determined by binning the cohort's PDX samples into passages according to the passage number and counting the number of bins, $N_{pb}$. This "collapse" of passages is designed to avoid a type of overrepresentation that would result from model expansion. A score of $S_{arm} = 1$ indicates that for each passage bin $N_{pb}$, there is a PDX sample (possibly more than 1) with the represented passage number that is a positive match for the study arm target. For $S_{arm} < 1$, there will be a passage number across all of the cohort's PDX model lines for which no samples match the study arm target. $S_{arm}$ and $N_{pb}$ together provide a measure of the depth of targetable passages in a PDX model cohort. The significance of this combination is to indicate which cohorts may be more amenable to drug studies across multiple passage numbers ($S_{arm} \sim 1$, especially with large $N_{pb}$) versus those that may not ($S_{arm} \ll 1$ or small $N_{pb}$). We calculated the percentage of unique passages that display positive signals for the target arms and show 19 arms with $S_{arm} = 1$ (Supplementary Fig. 6b) that may be conducive to drug trials.

**Reporting summary**. Further information on research design is available in the Nature Research Reporting Summary linked to this article.

## Data availability

Clinical and sequence data were obtained from the NIH-NCI PDX Development and Trial Centers Research Network (PDXNet) Consortium (https://www.pdxnetwork.org) and from the National Cancer Institute (NCI) Patient-Derived Models Repository (PDMR) public site (https://pdmr.cancer.gov). The PDXNet PDX Data Commons and Coordinating Center (PDCCC) curated data from the Washington University PDX Development and Trials Center (WU-PDTC), the University of Texas MD Anderson Cancer Center (MDACC), Huntsman Cancer Institute (HCI), The Wistar Institute (WI), and Baylor College of Medicine (BCM) and made them available on Seven Bridges' Cancer Genomics Cloud (The Cancer Genomics Cloud[90], https://cgc.sbgenomics.com) under the PDXNet Data Sharing agreement (i.e. the current policy mechanism for data release and sharing instituted by the PDXNet Data Coordination Center). Hyperlinks to these PDX centers and model descriptions are listed in Supplementary Data 6. Sequence data from PDXNet are being shared as part of the NCI Cancer Moonshot Initiative through the NCI Cancer Data Service (https://datacommons.cancer.gov/repository/cancer-data-service), under the mechanism used by a recent study of copy number profiles in PDXs[91]. Data from individual models can also be accessed publicly via https://portal.pdxnetwork.org. For materials that are subject to dbGaP restrictions, such as raw sequence data, information is provided on the portal site for how to access it. Omics results, which include somatic mutations, copy number segment-level and gene-level, copy number chromosome arm-level, fusion, and gene expression data, have been deposited as compressed, tabular plain-text files at Figshare (https://doi.org/10.6084/m9.figshare.14390408) and have been reformatted for viewing through the PDX Variant Viewer web portal (https://pdx.wustl.edu/pdx). Published datasets used in our analysis and their web sites are as follows: GENCODE (https://www.gencodegenes.org); COSMIC (https://cancer.sanger.ac.uk/cosmic); TCGA-MC3 (https://gdc.cancer.gov/about-data/publications/mc3-2017); GDC panel-of-normals (PON) (https://gdc.cancer.gov/about-data/gdc-data-processing/gdc-reference-files); gnomAD (https://gnomad.broadinstitute.org); dbSNP (https://ftp.ncbi.nih.gov/snp/organisms/human_9606_b151_GRCh38p7); dbNSFP (https://sites.google.com/site/jpopgen/dbNSFP); CIViC (https://civicdb.org); DEPO (https://github.com/ding-lab/publicDEPO); NCI-MATCH/EAY131 Precision Medicine Trial (https://ecog-acrin.org/trials/nci-match-eay131). Source data are provided with this paper.

## Code availability

The code for data processing that support these findings is available from the GitHub repository https://github.com/ding-lab/PDX-PanCanAtlas (Zenodo https://doi.org/10.5281/zenodo.4676237), and the SeqQEst codes are available on GitHub (https://github.com/ding-lab/SeqQEst).

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

## Acknowledgements

This project has been funded by the National Cancer Institute under award U54-CA224083 to L.D, S.L., and R.G. Additional support was provided by The Foundation for Barnes-Jewish Hospital's Cancer Frontier Fund through the Siteman Cancer Center Investment Program. The breast cancer PDX models from Washington University in St. Louis were developed in part through support from The Breast Cancer Research Foundation and Fashion Footwear Charitable Foundation of New York, Inc. PDMR data were generated with funding from the National Cancer Institute (Contract Number HHSN261200800001E). PDX generation and whole exome sequencing at the University of Texas MD Anderson Cancer Center were supported by the University of Texas MD Anderson Cancer Center Moon Shots Program, Specialized Program of Research Excellence (SPORE) grant CA070907. The development of PDX models and generation of data from Wistar Institute were supported by National Cancer Institute, National Institutes of Health (NCI R50-CA211199). Sample procurement and next-generation sequencing at Huntsman Cancer Institute was performed at the Genomics and Bioinformatics Analysis and Biorepository and Molecular Pathology Shared Resources, respectively, supported by NCI P30CA042014. M.T.L. is supported by a P30 Cancer Center Support Grant CA125123 and a Core Facility Support Grant from the Cancer Research and Prevention Initiative of Texas RP170691. Support for the PDXNET consortium included funding provided by the NIH to the PDXNet Data Commons and Coordination Center (NCI U24-CA224067), to the PDX Development and Trial Centers (NCI U54-CA224083, NCI U54-CA224070, NCI U54-CA224065, NCI U54-CA224076, NCI U54-CA233223, and NCI U54-CA233306). The Seven Bridges Cancer Research Data Commons Cloud Resource has been funded in whole or in part with Federal funds from the National Cancer Institute, National Institutes of Health, Contract No. HHSN261201400008C and ID/IQ Agreement No. 17 × 146 under Contract No.
HHSN261201500003I and 75N91019D00024. The Jackson Laboratory (JAX) PDX resource data were supported by the National Cancer Institute under the JAX Cancer Center NCI Grant (Award Number P30CA034196). The genomic data for JAX PDX tumors used in this work were generated by JAX Genome Technologies and Single Cell Biology Scientific Service. Finally, this project would not have been possible without the generous donation of tissues by our patients.

## Author contributions

L.D. led project design, study conception. R.J.M. collected data and developed PDX database and web portal. H.S. led pipeline development, performed data processing and data analysis, generated figures and tables, wrote the manuscript. H.S., S.C., and L.D. contributed to discuss mouse contamination filter approaches. S.C., R.J.M., C.K.M., S.Z. performed data analysis, generated figures and tables, wrote the manuscript. M.C.W. reviewed and edited the manuscript. M.A.W. generated figures. F.M.R., N.V.T., and Y.L. analyzed data and wrote methods. S.R.D., M.H.B., T.M.P., J.H., J.L.M., D.A.D.II4, R.P., L.C., R.J., K.L., A.W., B.A.V.T., C.X.M., R.A., K.F., J.F.D., The NCI PDXNet Consortium, B.D., M.T.L., M.D., M.H., B.F., J.A.R., A.L.W., B.E.W., F.M.B., F.C., R.C.F., S.L., R.G., J.H.D., J.A.M., Y.A.E., J.H.C., B.J.R., L.D. contributed sample collection, sequencing data generation, reviewed and edited the manuscript.

## Competing interests

The University of Utah may choose to license PDX models developed in the Welm labs, which may result in tangible property royalties to Drs. Welm and members of their labs who developed the models. M.T.L. is a founder and limited partner in StemMed Ltd. and a manager in StemMed Holdings, its general partner. He is a founder and equity stakeholder in Tvardi Theraeutics Inc. Some PDXs are exclusively licensed to StemMed Ltd. resulting in royalty income to M.T.L. L.E.D. is a compensated employee of StemMed Ltd. The other authors declare no competing interests.

## Consents

All patients were consented to the publishing of their de-identified clinical information. For further details, see Supplementary Methods.

## Ethics

All of the xenograft studies were completed in accordance with animal research ethics regulations of each PDTC's respective institutional review board. For further details, see the Supplementary Methods.

## Additional information

[1]Department of Medicine, Washington University in St. Louis, St. Louis, MO, USA. [2]McDonnell Genome Institute, Washington University in St. Louis, St. Louis, MO, USA. [3]Department of Computer Science, Princeton University, Princeton, NJ, USA. [4]Computational Cancer Genomics Research Group and Cancer Research UK Lung Cancer Centre of Excellence, University College London Cancer Institute, London, UK. [5]Department of Mathematics, Washington University in St. Louis, St. Louis, MO, USA. [6]Department of Genetics, Washington University in St. Louis, St. Louis, MO, USA. [7]Huntsman Cancer Institute, University of Utah, Salt Lake City, UT, USA. [8]Seven Bridges Genomics, Inc., Cambridge, Charlestown, MA, USA. [9]Frederick National Laboratory for Cancer Research, Frederick, MD, USA. [10]Siteman Cancer Center, Washington University in St. Louis, St. Louis, MO, USA. [11]Department of Radiation Oncology, Washington University in St. Louis, St. Louis, MO, USA. [12]Department of Otolaryngology, Washington University St. Louis, St. Louis, MO, USA. [13]Lester and Sue Smith Breast Center, Baylor College of Medicine, Houston, TX, USA. [14]The University of Texas MD Anderson Cancer Center, Houston, TX, USA. [15]The Wistar Institute, Philadelphia, PA, USA. [16]Division of Cancer Treatment and Diagnosis, National Cancer Institute, Bethesda, MD, USA. [17]Investigational Drug Branch, National Cancer Institute, Bethesda, MD, USA. [18]The Jackson Laboratory for Genomic Medicine, Farmington, CT, USA. [27]These authors contributed equally: Hua Sun, Song Cao, R. Jay Mashl, Chia-Kuei Mo, Simone Zaccaria. *A list of authors and their affiliations appears at the end of the paper. ✉email: lding@wustl.edu

## The NCI PDXNet Consortium

Li Ding[1,2,6,10]✉, Ramaswamy Govindan[1,10], Shunqiang Li[1,10], Rebecca Aft[10], Julie Belmar[1], Song Cao[1,2,35], Feng Chen[1], Sherri R. Davies[1], John F. Dipersio[1,10], Ryan C. Fields[10], Katherine C. Fuh[1,10], Jason Held[1], Jeremy Hoog[1], Reyka G. Jayasinghe[1,2], Yize Li[1,2], Kian-Huat Lim[1,10], Jingqin Luo[10], Cynthia X. Ma[1,10], R. Jay Mashl[1,2,35], Chia-Kuei Mo[1,2,35], Jacqueline L. Mudd[1], Fernanda Martins Rodrigues[1,2], Hua Sun[1,2,35], Nadezhda V. Terekhanova[1,2], Brian A. Van Tine[1,10], Rose Tipton[1], Andrea Wang-Gillam[1,10], Michael C. Wendl[2,5,6], Matthew A. Wyczalkowski[1,2], Yige Wu[1,2], Lijun Yao[1,2], Daniel Cui Zhou[1,2], Alana L. Welm[7], Bryan E. Welm[7], Matthew H. Bailey[7], Andrew Butterfield[7], Zhengtao Chu[7], Maihi Fujita[7], Chieh-Hsiang Yang[7], Emilio Cortes-Sanchez[7], Sandra Scherer[7], Ling Zhao[7], Tijana Borovski[8], Vicki Chin[8], Brandi Davis-Dusenbery[8], Dennis A. Dean II[8], John DiGiovanna[8], Christian Frech[8], Jeffrey Grover[8], Ryan Jeon[8], Soner Koc[8], Jelena Randjelovic[8], Sara Seepo[8], Tamara Stankovic[8], Yvonne A. Evrard[9], Rajesh Patidar[9], Li Chen[9], Michael T. Lewis[13], Lacey E. Dobrolecki[13], Matthew J. Ellis[13], Michael Ittmann[19], Susan G. Hilsenbeck[13], Bert W. O'Malley[20], Nicholas Mitsiades[20,21], Salma Kaochar[21], Jack A. Roth[14], Funda Meric-Bernstam[14], Michael A. Davies[14], Argun Akcakanat[14], Jithesh Augustine[14], Huiqin Chen[14], Bingbing Dai[14], Kurt W. Evans[14], Bingliang Fang[14], Kelly Gale[14], Don Gibbons[14], Min Jin Ha[14], Vanessa Jensen[14], Michael Kim[14], Bryce P. Kirby[14], Scott Kopetz[14], Christopher D. Lanier[14], Dali Li[14], Mourad Majidi[14], David Menter[14], Ismail Meraz[14], Turcin Saridogan[14], Stephen Scott[14], Alexey Sorokin[14], Coya Tapia[14], Jing Wang[14], Shannon Westin[14], Yuanxin Xi[14], Yi Xu[14], Fei Yang[14], Timothy A. Yap[14], Vashisht G. Yennu-Nanda[14], Erkan Yuca[14], Jianhua Zhang[14], Ran Zhang[14], Xiaoshan Zhang[14], Xiaofeng Zheng[14], Meenhard Herlyn[15], Dylan Fingerman[15], Haiyin Lin[15], Qin Liu[15], Andrew V. Kossenkov[15], Vito W. Rebecca[15], Rajasekharan Somasundaram[15], Michae T. Tetzlaff[22], Jayamanna Wickramasinghe[15], Min Xiao[15], Xiaowei Xu[22], James H. Doroshow[16], Jeffrey A. Moscow[17], Jeffrey H. Chuang[18], Carol J. Bult[18], Peter N. Robinson[18], Anuj Srivastava[18], Michael W. Lloyd[18], Steven B. Neuhauser[18], Jill Rubinstein[18], Brian J. Sanderson[18], Brian White[18], Xing Yi Woo[18], Tiffany Wallace[22], John D. Minna[23], Gao Boning[23], Luc Girard[23], Hyunsil Park[23],

Brenda C. Timmons[23], Katherine L. Nathanson[24], George Xu[25], Chong-xian Pan[26], Moon S. Chen Jr[26], Luis G. Carvajal-Carmona[26], May Cho[26], Nicole B. Coggins[26], Ralph W. deVere White[26], Guadalupe Polanco-Echeverry[26], Ana Estrada[26], David R. Gandara[26], Amanda R. Kirane[26], Tiffany Le[26], Paul Lott[26], Alexa Morales Arana[26], Jonathan W. Reiss[26], Sienna Rocha[26], Clifford G. Tepper[26], Ted Toal[26], Hongyong Zhang[26] & Ai-Hong Ma[26]

[19]Department of Pathology, Baylor College of Medicine, Houston, TX, USA. [20]Department of Molecular and Cellular Biology, Baylor College of Medicine, Houston, TX, USA. [21]Department of Medicine, Baylor College of Medicine, Houston, TX, USA. [22]Center to Reduce Cancer Health Disparities, National Cancer Institute, Bethesda, MD, USA. [23]Hamon Center for Therapeutic Oncology, UT Southwestern Medical Center, Dallas, TX, USA. [24]Abramson Cancer Center, University of Pennsylvania, Philadelphia, PA, USA. [25]Department of Pathology and Laboratory Medicine, Hospital of the University of Pennsylvania, Philadelphia, PA, USA. [26]University of California Davis, Sacramento, CA, USA.

