## [Peer Review File · Nature Communications]

REVIEWERS' COMMENTS

Reviewer #1 (Remarks to the Author):

The authors have well addressed my comments.

I would suggest including the draggable target comparison in point 2 as a supplementary figure (space permitting) to complement the NCI-MATCH data as this is a common consideration in early drug development and would further encourage the use of PDXs.

Andrew Biankin

Reviewer #2 (Remarks to the Author):

Reviewer#2 (remark to the authors)

The reviewer appreciates the effort made in answering the number of questions raised about the samples used in a study of this complexity. The authors have carried out a significant number of bioinformatic studies answering the questions raised. They have also improved the presentation of the data of the models that are the object of study, facilitating their traceability and answering important questions about the quality of biological samples. They have responded convincingly to most of the points suggested by the reviewer. After reviewing the article again with the changes made to the text and figures, I have no major scientific concerns.

Reviewer #3 (Remarks to the Author):

I commend the authors on their detailed response to the reviewers' comments. They have addressed my comments adequately except for 2 points:

- 1) The impact of the work is greatly diminished by the lack of pharmacological profiles for the PDXs investigated at the molecular level. Providing a resource to the community where both molecular and pharmacological data are available would be highly beneficial to the scientific community
- 2) Ensuring full research reproducibility of the data processing, curation and analysis should be possible but this was achieved for the authors' study. This is mitigated by the fact that part of the code is being shared and that the authors developed a basic web-application to explore the data.

REVIEWERS' COMMENTS

Reviewer #1 (Remarks to the Author):

The authors have well addressed my comments.

I would suggest including the draggable target comparison in point 2 as a supplementary figure (space permitting) to complement the NCI-MATCH data as this is a common consideration in early drug development and would further encourage the use of PDXs.

Andrew Biankin

Authors: Thank you for the reviewer's suggestion. We take the referee's comment. In order to comply with the logic of the article, we reanalyzed the point 2 and added it to the supplementary figure in the revision (Supplementary Figure 6a).

Reviewer #2 (Remarks to the Author):

The reviewer appreciates the effort made in answering the number of questions raised about the samples used in a study of this complexity. The authors have carried out a significant number of bioinformatic studies answering the questions raised. They have also improved the presentation of the data of the models that are the object of study, facilitating their traceability and answering important questions about the quality of biological samples. They have responded convincingly to most of the points suggested by the reviewer. After reviewing the article again with the changes made to the text and figures, I have no major scientific concerns.

Authors: We appreciate the referee's thorough consideration of the revision and are delighted that it has satisfied the concerns.

Reviewer #3 (Remarks to the Author):

I commend the authors on their detailed response to the reviewers' comments. They have addressed my comments adequately except for 2 points:

1) The impact of the work is greatly diminished by the lack of pharmacological profiles for the PDXs investigated at the molecular level. Providing a resource to the community where both molecular and pharmacological data are available would be highly beneficial to the scientific community

Authors: Thank you for your comments on our work. We provided the molecular results and pharmacological data in the manuscript. The molecular results are available Supplementary Tables and Figshare website (free download) doi.org/10.6084/m9.figshare.13050512, and pharmacological data is available in Supplementary Data 3 and 5.

2) Ensuring full research reproducibility of the data processing, curation and analysis should be possible but this was achieved for the authors' study. This is mitigated by the fact that part of the code is being shared and that the authors developed a basic web-application to explore the data.

Authors: We share the referee's philosophy pertaining to transparency and reproducibility and feel that code availability, and especially the data portal, will be useful tools for the community with regards to the results of this study.